# Loss of PHF6 causes spontaneous seizures, enlarged brain ventricles and altered transcription in the cortex of a mouse model of the Börjeson–Forssman–Lehmann intellectual disability syndrome

Helen M. McRae[1,2°], Melody P. Y. Leong[1,2°], Maria I. Bergamasco[1,2], Alexandra L. Garnham[1,2], Yifang Hu[1], Mark A. Corbett[3], Lachlan Whitehead[1,2], Farrah El-Saafin[1,2], Bilal N. Sheikh[1,2], Stephen Wilcox[1,2], Anthony J. Hannan[4,5], Jozef Gécz[3,6], Gordon K. Smyth[1,7], Tim Thomas[1,2‡]*, Anne K. Voss[1,2‡]*

1 Walter and Eliza Hall Institute of Medical Research, Melbourne, Victoria, Australia, 2 Department of Medical Biology, The University of Melbourne, Parkville, Victoria, Australia, 3 Robinson Research Institute and Adelaide Medical School, The University of Adelaide, Adelaide, South Australia, Australia, 4 Florey Institute of Neuroscience and Mental Health, University of Melbourne, Parkville, Victoria, Australia, 5 Department of Anatomy and Physiology, University of Melbourne, Parkville, Victoria, Australia, 6 South Australian Health and Medical Research Institute, Adelaide, South Australia, Australia, 7 School of Mathematics and Statistics, The University of Melbourne, Parkville, Victoria, Australia

° These authors contributed equally to this work.
‡ These authors are joint senior authors on this work.
* tthomas@wehi.edu.au (TT); avoss@wehi.edu.au (AKV)

**Data Availability Statement:** Lists of all processed RNA-sequencing results, including all expressed

## Abstract

Börjeson-Forssman-Lehmann syndrome (BFLS) is an X-linked intellectual disability and endocrine disorder caused by pathogenic variants of plant homeodomain finger gene 6 (*PHF6*). An understanding of the role of PHF6 *in vivo* in the development of the mammalian nervous system is required to advance our knowledge of how *PHF6* mutations cause BFLS. Here, we show that PHF6 protein levels are greatly reduced in cells derived from a subset of patients with BFLS. We report the phenotypic, anatomical, cellular and molecular characterization of the brain in males and females in two mouse models of BFLS, namely loss of *Phf6* in the germline and nervous system-specific deletion of *Phf6*. We show that loss of PHF6 resulted in spontaneous seizures occurring via a neural intrinsic mechanism. Histological and morphological analysis revealed a significant enlargement of the lateral ventricles in adult *Phf6*-deficient mice, while other brain structures and cortical lamination were normal. *Phf6* deficient neural precursor cells showed a reduced capacity for self-renewal and increased differentiation into neurons. *Phf6* deficient cortical neurons commenced spontaneous neuronal activity prematurely suggesting precocious neuronal maturation. We show that loss of PHF6 in the foetal cortex and isolated cortical neurons predominantly caused upregulation of genes, including *Reln*, *Nr4a2*, *Slc12a5*, *Phip* and ZIC family transcription factor genes, involved in neural development and function, providing insight into the molecular effects of loss of PHF6 in the developing brain.

genes, differentially expressed genes, enriched GO terms and KEGGS pathways are available as S3 and S4 Tables. RNA-sequencing raw data files are available under GEO accession number GSE275729. The source data underlying Figs 1 to 5 and 7 and S1, S2, S4, S6, S7, S8, S10, S11, S14 and S15 are provided as a source data file S1 Data.

**Funding:** This work was supported by an Australian Postgraduate Award (H.M.M.), National Health and Medical Research Council (NHMRC) Project grants (1029481 and 1084248 to A.K.V.), NHMRC Research Fellowships (1003435 [T.T.], 575512 and 1081421 [A.K.V.], 1154970 [G.K.S.] and 1155224 [J.G.], NHMRC investigator grants 1176789 [A.K.V.] and 2025645 [G.K.S.]), Independent Research Institutes Infrastructure Support Scheme from the Australian Government's NHMRC, and a Victorian State Government Operational Infrastructure Support Grant. The funders had no role in study design, data collection and analysis, decision to publish, or preparation of the manuscript.

**Competing interests:** The authors have declared that no competing interests exist.

## Author summary

The Börjeson-Forssman-Lehmann Syndrome (BFLS) is an intellectual disability and endocrine disorder. Mutations in the plant homeodomain finger 6 gene (PHF6) cause the disorder. We show here that a subset of BFLS patients lack PHF6 and report the effects of loss of PHF6 in an animal model with complete loss of PHF6 from conception. The cerebral cortex is the site of higher brain functions, including cognition and decision-making. We report here the effects of lack of PHF6 on the developing brain, cerebral cortex and neuronal cells isolated from this structure. Since PHF6 associates with the genetic material in the cell nucleus and has been proposed to regulate gene activity, we also report the effects of lack of PHF6 on gene expression in the cerebral cortex and purified neuronal cells. We observed that loss of PHF6 results in the dysregulation of neuronal development and differentiation genes, including genes involved in disorders such as Parkinson's disease, epilepsy, neuroblastoma, attention deficit disorder, autism and schizophrenia. Lastly, we report that mice lacking PHF6 mirror BFLS patients in that they also suffer from spontaneous epileptic seizures. Our mouse models and findings will be useful to further investigate BFLS neuronal function and other aspects of the disorder in the adult.

## Introduction

Börjeson–Forssman–Lehmann syndrome (OMIM: 301900) is a rare X-linked intellectual disability syndrome, first described in 1962 [1] and caused by inherited or *de novo* mutations in the X-linked *PHF6* gene [2]. Males hemizygous for *PHF6* mutations are invariably affected, while females heterozygous for *PHF6* mutations range from unaffected carriers to a full presentation of the syndrome [3,4]. Individuals with BFLS generally present with mild to severe intellectual disability and a number of defining physical characteristics including short stature, small genitalia, hypogonadism, gynecomastia, large earlobes, deep-set eyes, long tapered fingers and foreshortened toes [5,6]. Seizures occur in 8 to 29% of individuals with confirmed *PHF6* mutations [7–9]. Mutations in *PHF6* that cause BFLS include nonsense and frameshift mutations, multi-exon and whole gene deletions and occur along the length of the protein, implicating a loss or reduced function mechanism [9,10]. Somatic *PHF6* mutations also occur in blood cancers, most frequently in T-cell acute lymphoblastic leukaemia [11]. PHF6 plays a role in haematopoietic stem and progenitor cell homeostasis [12,13] and in the negative regulation of interferon signalling [12]. PHF6 is a haematological tumour suppressor [12]. Indeed, two of the approximately 50 described males with BFLS have developed a haematological malignancy to date [14,15]. Mice lacking PHF6 from conception develop to birth on a C57BL/6 inbred genetic background [12]. On an outbred background they reach adulthood but display defects in growth hormone signalling and a reduction in body size [16], mirroring the short stature observed in BFLS individuals.

The PHF6 protein contains four nuclear localization sequences and two atypical plant homeodomain (PHD) zinc-fingers, similar to the imperfect PHD of the mixed lineage leukaemia protein [2]. The mouse and human PHF6 proteins are 97.5% identical [17]. Canonical PHD zinc-fingers regulate transcription through binding of modified or unmodified histone tails [18–20]. PHF6 has been reported to associate with double-stranded DNA [21] and histone variants including H3 in B-cell leukaemia cells [22] and H3.1, H2B.1, H2A.Z, and H1.2 in HEK293T cells [23]. PHF6 has been proposed to bind H2B acetylated on lysine 12 and to act

as an E3 ubiquitin ligase for the ubiquitination of H2B on lysine 120 during embryonic stem cell differentiation [24]. PHF6 has been reported to interact with the nucleosome remodelling complexes, including NuRD, ISWI and SWI/SNF in Jurkat and HEK293T cells [23,25], as well as the PAF1 transcriptional elongation complex in primary rat cortical neurons [26]. Localization of PHF6 to the nucleolus has also been reported [2], and PHF6 has been shown to interact with UBF [27], ribosomal proteins, splicing factors [23] and has been proposed to negatively regulate rRNA transcription in HeLa and HEK293T cells [23,27]. Thus, PHF6 functions in the nucleus and nucleolus and mediates interactions with chromatin.

Few neuroanatomical studies have been reported for BFLS. Some common findings include dilated ventricles, which were identified in three of five male patients in one study (assessed by CT scan) [28] and in two of three females with heterozygous *PHF6* mutations as assessed by MRI [4,7]. Abnormalities of the cerebral cortex have also been reported, including in the original BFLS family where a paucity of certain cell types, pyknotic nuclei and gliosis was observed in the cortex in one patient upon autopsy [1]. Areas of undetectable lamination, especially in the superficial granular layer were noted in another post-mortem male patient brain, in addition to heterotopic neurons in the cerebral cortex, subcortical white matter and surrounding the ventricular system, and heterotopic grey matter found among the white matter [29]. In one female with BFLS, increased signalling of the white matter was noted on an MRI [7], while grey matter heterotopias were identified in two other females with BFLS [30]. In summary, abnormalities of the lateral ventricles and cerebral cortex have been described in a subset of cases, however, given the scarcity of reported patient studies, the neuropathological features of BFLS are not well defined.

*Phf6* mRNA and protein are present in all cells in mice [17]. They are more abundant during embryonic and foetal development and then decline in the postnatal period. *Phf6* mRNA is present at low levels in adult tissues [17]. PHF6 protein is detectable in adult tissues using a high-affinity antibody [12]. Complete loss of PHF6 in mice on a C57BL/6 inbred background results in perinatal lethality [12,31], but heterozygous females survive [12]. A mouse model of BFLS using a patient-specific hypomorphic mutation (replacement of cysteine 99 of the protein with phenylalanine, *Phf6*$^{C99F}$) recapitulated a number of features of BFLS including a smaller body size and cognitive impairment [31]. In response to the GABA antagonist pentylenetetrazol, *Phf6*$^{C99F}$ mice displayed a reduced latency to seizure induction, and after stimulation with glutamate agonist, kainic acid, increased seizure severity [31]. However, spontaneous seizures were not reported. Another mouse model of BFLS using a patient-specific hypomorphic mutation (*Phf6*$^{R342X}$) showed reduced brain volume and hydrocephaly [32]. A previous study using shRNA to knockdown *Phf6 in utero* in the developing cortex suggested that reducing PHF6 expression slowed neuronal migration [26], indicating that PHF6 may control cortical layering. In contrast, nervous-system specific deletion of *Phf6* prior to cortex development increased the depth of layer I, but did not affect the order of the layers [33], and a patient-specific mutation in PHF6 (*Phf6*$^{R342X}$) caused no cortical lamination defects [32]. Whether germline *Phf6* null mutation causes a cortical layering defect has not yet been tested. PHF6 has been shown to affect the transcription of a number of genes in the postnatal cerebral cortex, including promoting expression of cell differentiation genes and repressing potassium transport and synaptic genes [31], as well as promoting the expression of ephrin receptor genes [33]. Similarly, a number of genes were differentially expressed upon shRNA knockdown of *Phf6* in E16.5 rat cortical neurons [26]; however, the effect of complete germline deletion of *Phf6* on gene expression in the foetal cerebral cortex has not been established. To progress our understanding of how *PHF6* mutations cause BFLS, we studied the impact of germline deletion and nervous system-specific deletion of *Phf6* in both male and female mice.

## Results

### Effect of *PHF6* mutations in BFLS on PHF6 protein levels

To examine the effect of human *PHF6* mutations on PHF6 protein levels, we isolated protein from cell lines established from peripheral blood lymphoblasts of patients with BFLS [2,34,35] and controls (Fig 1A and S1 Table). Certain *PHF6* mutations showed dramatically reduced protein level (p.C45Y, p.C99F), while the recurrent p.R342* mutation [2,36] produced a low amount of truncated PHF6 protein. Other mutations (p.D333del, p.E337del) showed PHF6 protein levels similar or even elevated compared to controls. These results demonstrate that at least a subset of human *PHF6* mutations in BFLS cause reduced PHF6 protein.

### Effects of the *Phf6* exons 4 and 5 deletion on PHF6 protein levels in mice

Our western blot data from individuals with BFLS suggested that an animal model with a *Phf6* allele resulting in loss of PHF6 protein would be suitable to model aspects of the disorder. We flanked exons 4 and 5 of the mouse *Phf6* gene with *loxP* sites, which after recombination results in complete loss of PHF6 protein [12]. *PHF6* exon 4 and 5 deletion has been reported for one BFLS individual [9].

Deletion of *Phf6* exons 4 and 5 resulted in the absence of RNA sequencing reads mapping to these exons in embryonic day 15.5 (E15.5) mouse cerebral cortex and in cortical neurons isolated from E16.5 mouse foetuses (Fig 1B and 1C) and the absence of PHF6 protein in E15.5 mouse brain (Fig 1D).

### Loss of PHF6 causes spontaneous seizures through a nervous system intrinsic mechanism

As previously reported [12], males with *Phf6* germline deletion died perinatally on a C57BL/6 background; however, females with heterozygous *Phf6* mutation were present at the expected ratio at weaning (S2 Table). We monitored a group of females heterozygous for the *Phf6* mutation and found that seizures occurred in $Phf6^{+/-}$ mice from 172 days of age onwards with a median latency of 622 days (p < 0.0001, Fig 2A). The observed seizures were generalised tonic-clonic seizures with falling, stage 5 according to Racine [37]. To test whether seizures occurred via a neural-intrinsic mechanism, we employed the *Nestin-cre* transgene [38] to delete *Phf6* in the neural tube, which is active from E11.5 [39]. In females with neural-specific deletion of one copy of *Phf6* ($Phf6^{+/lox}$; $Nes-cre^{Tg/+}$), we observed a median latency of seizure onset of 600 days (p = 0.03, Fig 2B). Similarly, males with hemizygous *Phf6* deletion in the nervous system ($Phf6^{lox/Y}$;$Nes-cre^{Tg/+}$) developed seizures with a median latency of 600 days (p = 0.002, Fig 2C). Deletion of both copies of *Phf6* in the nervous system in a small number of female mice ($Phf6^{lox/lox}$;$Nes-cre^{Tg/+}$) also resulted in seizures (S1 Fig).

### Loss of PHF6 does not affect the number of GAD67[+] interneurons in the cerebral cortex

Interneurons mediate synaptic inhibition and interneuron reduction has been associated with seizures in mouse models such as the *Dlx1* knockout mice [40]. Therefore we assessed the number of GABA[+] interneurons using the marker GAD67 (glutamate decarboxylase, 67 kDa form) [41]. We found no effect of loss of PHF6 on the number of GAD67[+] inhibitory neurons in adult mice (95.98 ± 6.8 cells in $Phf6^{+/Y}$;$Nes-cre^{Tg/+}$ controls vs. 103.96 ± 8.8 cells in $Phf6^{lox/Y}$; $Nes-cre^{Tg/+}$ brains per mm$^2$ of cortex section; S2 Fig).

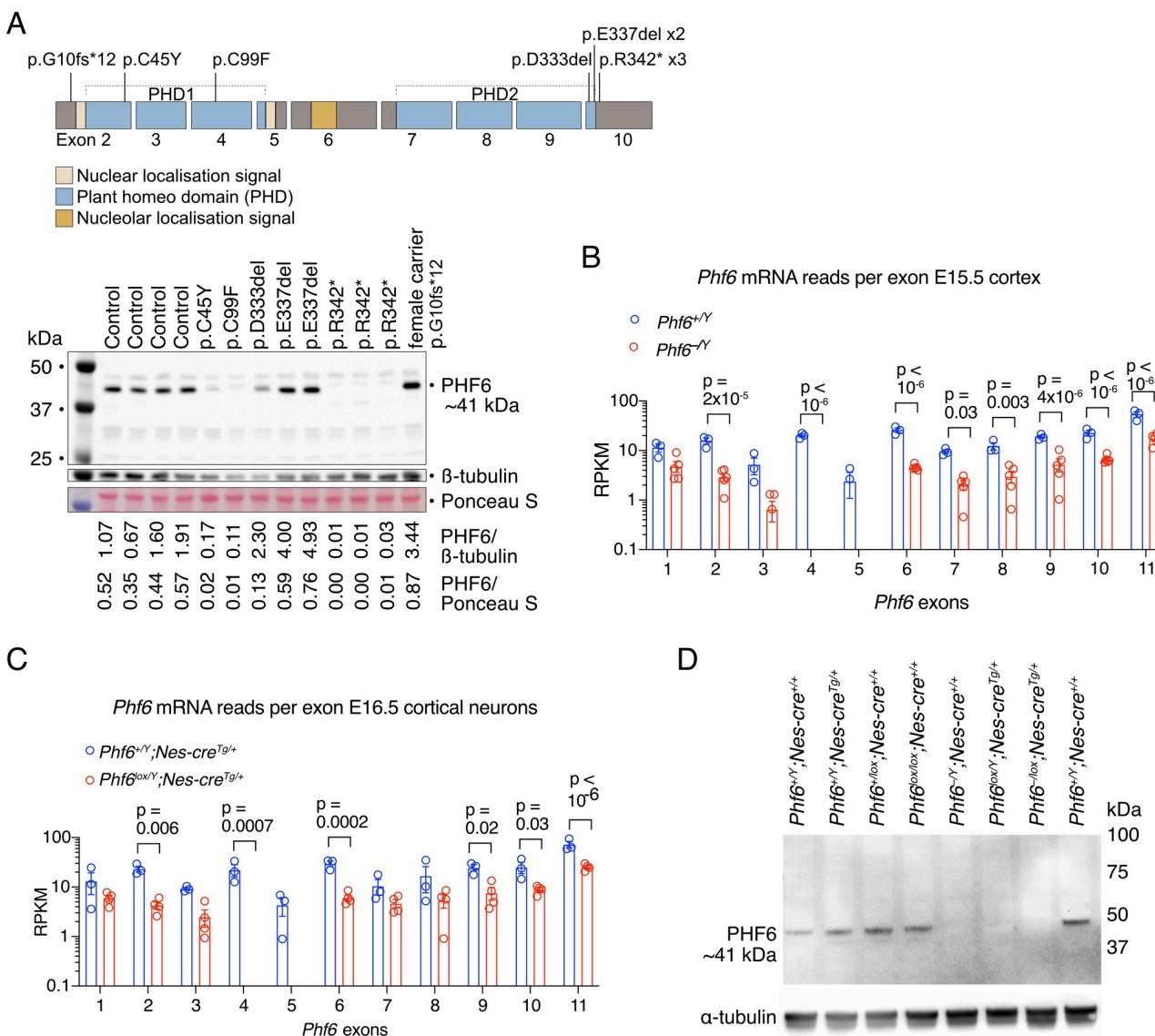

**Fig 1. PHF6 protein levels in cells from BFLS individuals with *PHF6* mutations, mRNA and protein levels in the cortex and cortical neurons of *Phf6* deleted mice.** (A) PHF6 protein expression is reduced in cell lines derived from lymphocytes of individuals affected by BFLS. Schematic drawing of the protein domains and coding exon structure of the *PHF6* gene with BFLS variants indicated and Western blot showing expression of PHF6 migrating at approximately 41 kDa in 8 males affected by BFLS, one female carrier and four unaffected males (control). Protein expression in individuals with the PHF6 p.C45Y, p.C99F and p.R342* variants was reduced, while the p.D333del and p.E337del variants are expressed at a similar or even higher level to the four unaffected individuals (control). Note the band migrating at approximately 38.5 kDa in the patients with the p.R342* variant indicating that the truncated protein is likely to be expressed at a low abundance. Details on individuals included in this figure are in S1 Table. (B, C) RNA-sequencing reads as reads per kilobase per one million reads in the sequencing library (RPKM) mapping to *Phf6* exons in *Phf6*$^{+/Y}$ vs. *Phf6*$^{-/Y}$ mouse E15.5 developing cortex (B) and *Phf6*$^{+/Y}$;*Nes-cre*$^{Tg/+}$ vs. *Phf6*$^{lox/Y}$;*Nes-cre*$^{Tg/+}$ mouse E16.5 cortical neurons (C). (D) Detection of PHF6 protein by western immunoblotting in E15.5 mouse brains of *Phf6* genotypes as indicated. Each lane represents an individual foetus. Note the absence of PHF6 protein at ~41 kDa in foetuses with germline deletion (*Phf6*$^{-/Y}$;*Nes-cre*$^{+/+}$) or nervous system-specific deletion (*Phf6*$^{lox/Y}$;*Nes-cre*$^{Tg/+}$) of the single allele of *Phf6* in males or germline deletion of one allele and nervous system-specific deletion of the second allele in the female (*Phf6*$^{-/lox}$;*Nes-cre*$^{Tg/+}$). N = 8 individuals with BFLS, 1 BFLS carrier and 4 control individuals (A), 3 *Phf6*$^{+/Y}$ and 5 *Phf6*$^{-/Y}$ E15.5 mouse foetuses (B), 3 *Phf6*$^{+/Y}$;*Nes-cre*$^{Tg/+}$ and 4 *Phf6*$^{lox/Y}$;*Nes-cre*$^{Tg/+}$ E16.5 mouse foetuses (C) and 3 *Phf6* deleted and 5 control E15.5 mouse foetuses, genotypes indicated above the blot (D). Data are displayed as mean ± sem and individual foetuses are represented by circles (B,C). RNA sequencing data (B,C) were analysed as described in the methods section; p values displayed above bars, two-way ANOVA with Šídák's multiple comparisons test.

A

Time to first seizure in female with heterozgyous germline *Phf6* deletion

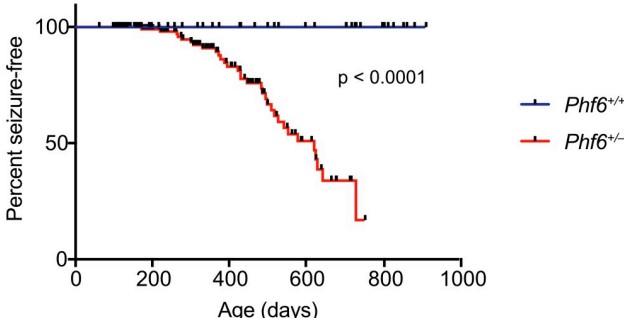

B

Time to first seizure in females with NS-specific *Phf6* heterozygous deletion

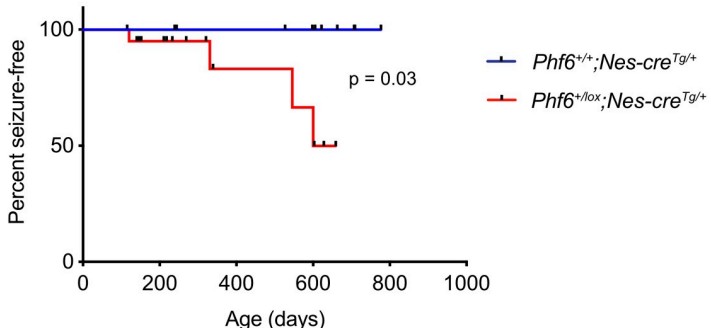

C   Time to first seizure in males with hemizygous NS-specific *Phf6* deletion

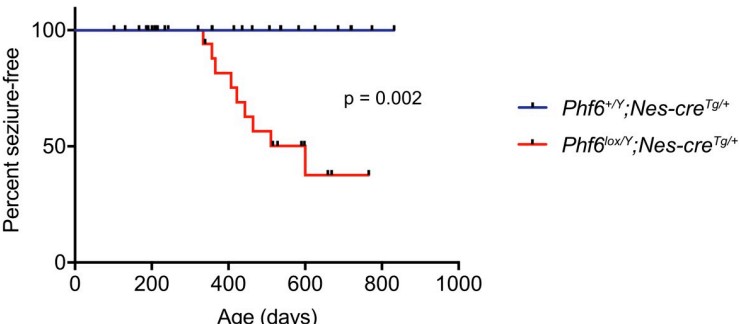

**Fig 2. Seizures occur spontaneously in mice with *Phf6* deletion through a neural intrinsic mechanism.** (A) Time to first seizure in female heterozygous germline deleted *Phf6*[+/−] mice compared to *Phf6*[+/+] wild-type control mice. Wild-type mice did not display seizures. Median time to seizure onset was 662 days. N = 70 *Phf6*[+/+] mice and 103 *Phf6*[+/−] mice. (B) Time to first seizure in females with *Phf6* heterozygous deletion in the nervous system compared to controls; *Phf6*[+/+];*Nestin-cre* transgenic control mice did not display seizures. N = 11 *Phf6*[+/+];*Nes-cre*[Tg/+] and 11 *Phf6*[lox/+];*Nes-cre*[Tg/+] mice. Time to seizure onset of a small number of *Phf6*[lox/lox];*Nes-cre*[Tg/+] mice (homozygous nervous system-specific deleted) is shown in S1 Fig. (C) Time to first seizure in males with *Phf6* deletion in the nervous system compared to controls. *Phf6*[+/Y];*Nestin-cre* transgenic control mice did not display seizures. N = 33 *Phf6*[+/Y];*Nes-cre*[Tg/+] and 13 *Phf6*[lox/Y];*Nes-cre*[Tg/+] mice. Checked points indicate mice that were sacrificed due to illness or other reasons. Data were analysed by a log-rank (Mantel-Cox) test. NS = nervous system. Survival data of mice with germline deletion of *Phf6* are shown in S2 Table.

## Effect of loss of PHF6 on brain anatomy and structure

To explore whether there were structural changes in the brains of mice that developed seizures, we prepared serial coronal cresyl-violet sections from three $Phf6^{lox/Y};Nes-cre^{Tg/+}$ brains 17 to 37 days after seizure onset and three age-matched controls and performed a section-by-section comparison (Fig 3A and S3 Fig). $Phf6$ is highly expressed in cerebellar Purkinje cells [17] and we therefore examined the Purkinje cells in the cerebellum of the $Phf6^{lox/Y};Nes-cre^{Tg/+}$ cresyl-violet-stained brains. We did not identify a difference in morphology or number of Purkinje cells in $Phf6^{lox/Y};Nes-cre^{Tg/+}$ vs. controls (30.2 ± 0.6 cells vs. 32.6 ± 1.3 cells per mm of Purkinje cell layer respectively, S4 Fig). Consistent with an overall reduced body size in $Phf6^{lox/Y};Nes-cre^{Tg/+}$ mice [16], $Phf6^{lox/Y};Nes-cre^{Tg/+}$ mice had a reduced brain volume within each pair (Fig 3B). In contrast, $Phf6^{lox/Y};Nes-cre^{Tg/+}$ mice had a larger lateral ventricle volume compared to controls (3.71 ± 0.17 mm$^3$ vs. 2.15 ± 0.51 mm$^3$, p = 0.04, Fig 3C). We quantified the volume of the cerebral cortex in the region surrounding the lateral ventricles and found that it was reduced in $Phf6^{lox/Y};Nes-cre^{Tg/+}$ brains (Fig 3D). Dividing the cerebral cortex volume by the total brain volume in the region surrounding the lateral ventricles showed that the cerebral cortex volume was disproportionally reduced relative to the total brain volume in $Phf6^{lox/Y};Nes-cre^{Tg/+}$ animals compared to controls (ratios are 0.308 ± 0.004 vs. 0.314 ± 0.003 for $Phf6$-deleted and controls, respectively; S5A Fig). The volume of the remaining areas of the $Phf6^{lox/Y};Nes-cre^{Tg/+}$ brains (i.e., brain volume minus cerebral cortex volume and lateral ventricle volume) relative to the total brain volume was also reduced compared to controls (ratios are 0.660 ± 0.003 vs. 0.671 ± 0.004 for $Phf6$-deleted and controls, respectively; S5B Fig). No other major abnormalities were observed. Lamination of the cerebral cortex (Fig 3E) and the relative depth of the cortical layers (S5C Fig) did not appear to be affected by loss of PHF6. To determine if the enlarged lateral ventricles preceded the onset of seizures or could potentially be a consequence of the seizures, we examined cresyl violet-stained serial sections of brains at 13 to 14 weeks of age, which was more than 10 weeks before the earliest spontaneous seizures were observed. At this stage preceding the onset of seizures, $Phf6^{lox/Y};Nes-cre^{Tg/+}$ already had enlarged ventricles compared to controls (p = 0.02 to 0.001; S6 Fig).

In summary, these data indicate that loss of PHF6 caused reduced brain size and an increase in ventricle volume.

## Incidence of hydrocephalus with enlargement of the skull is increased in mice with germline heterozygous *Phf6* mutation, but not nervous system specific deletion

An enlargement of lateral ventricles can be associated with hydrocephalus and enlargement of the skull, a condition that can occur spontaneously in between 1 to 3% mice on the C57BL/6 inbred background [42]. We observed an increased incidence of hydrocephalus with enlargement of the skull in $Phf6^{+/-}$ mice, with 11.5% of 205 monitored mice developing symptoms between 23 and 190 days, with the majority of cases occurring before 33 days. In contrast, no cases were observed in 268 $Phf6^{+/+}$ control mice (p < 0.0001, S7A Fig). On the other hand, in males and females with neural-specific mutation of $Phf6$, we did not observe an increased incidence of hydrocephalus (S7B and S7C Fig). Since an enlargement of the lateral ventricles, but not increased hydrocephalus, was identified after nervous system specific $Phf6$ deletion, this indicates that enlargement of the lateral ventricles associated with $Phf6$ mutation may predispose to but does not necessarily result in hydrocephalus with enlargement of the skull.

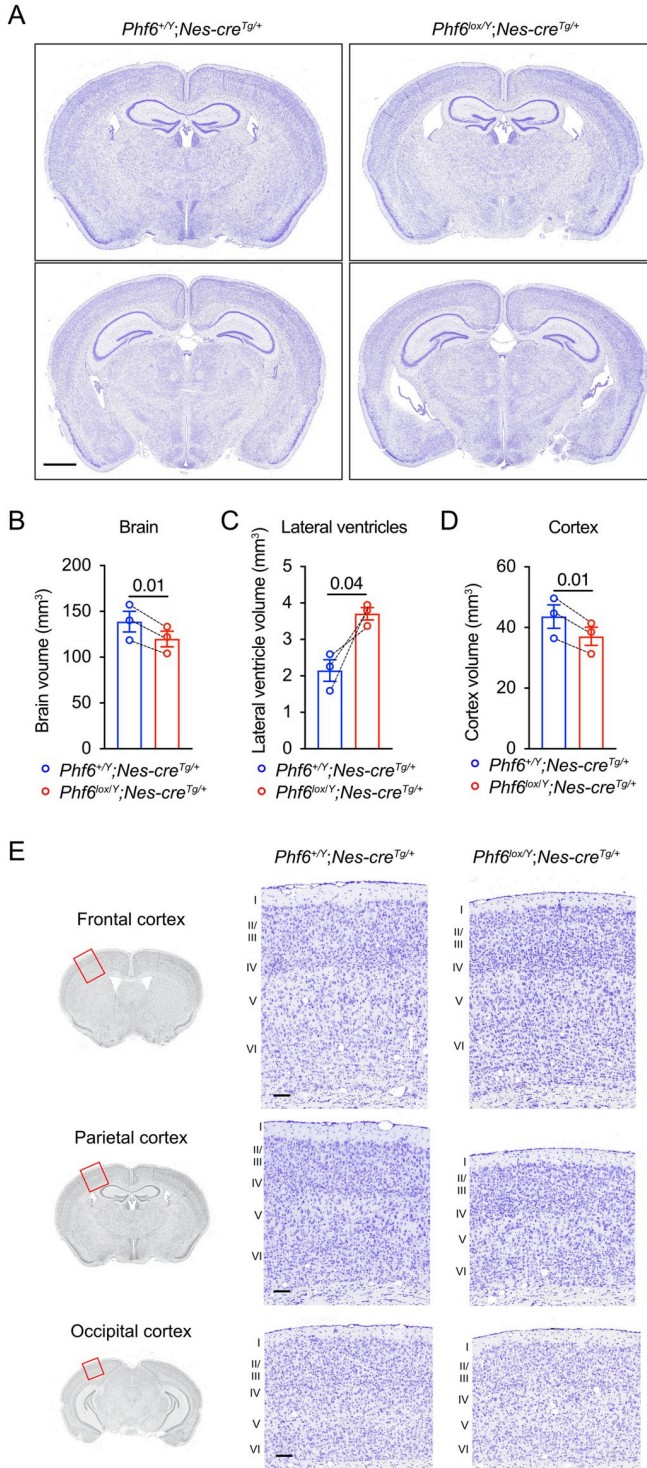

**Fig 3. Reduced brain size and enlarged lateral ventricles in adult *Phf6^{lox/Y};Nes-cre^{Tg/+}* mice.** (A) Images of 10 μm brain serial sections stained with cresyl-violet from a representative *Phf6^{lox/Y};Nes-cre^{Tg/+}* and a *Phf6^{+/Y};Nes-cre^{Tg/+}* paired control. Adult brains from 3 *Phf6^{lox/Y};Nes-cre^{Tg/+}* mice were fixed 17 to 37 days after onset of seizures, each with an age-matched *Phf6^{+/Y};Nes-cre^{Tg/+}* control (394 to 483 days old). Scale bar = 1 mm. (B-D) Volumes of the brain, lateral ventricles and cortex in *Phf6^{lox/Y};Nes-cre^{Tg/+}* versus *Phf6^{+/Y};Nes-cre^{Tg/+}* mice showing data here corresponding to the brain region spanning the lateral ventricles. Volumes were measured for the brain, lateral ventricle and cortex area in seven matched sections per animal across sections spaced at approximately 480 μm intervals. N = 3 age-matched pairs of *Phf6^{lox/Y};Nes-cre^{Tg/+}* mouse and *Phf6^{+/Y};Nes-cre^{Tg/+}* mice. Bars in the graph show the mean ± sem of

each genotype, while lines between individual data points join each $Phf6^{lox/Y}$;$Nes$-$cre^{Tg/+}$ animal to its paired control. Data were analysed by a two-tailed paired t-test. (E) Images showing representative cresyl-violet-stained sections of the cerebral cortex from the brains of $Phf6^{lox/Y}$;$Nes$-$cre^{Tg/+}$ and $Phf6^{+/Y}$;$Nes$-$cre^{Tg/+}$ mice. The approximate region of the cortex area shown is indicated by the red rectangles on the coronal images on the left. Numbers next the sections indicate the cortical layers I to VI. Scale bar = 100 μm. Related data are shown in S3, S5 and S6 Figs; assessment of interneuron number, Purkinje cell number and incidence of hydrocephalus are displayed in S2, S4 and S7 Figs, respectively.

## Loss of PHF6 does not affect cortical layering

Since the cerebral cortex, an important site of mammalian cognitive function, is often affected in intellectual disability, we focused our attention on defining the role of PHF6 in the cerebral cortex. The overall cell architecture of the cerebral cortex of $Phf6^{lox/Y}$;$Nes$-$cre^{Tg/+}$ brains stained with cresyl violet did not indicate any overt defects (Fig 3E and S5C Fig). However, given previous reports that knockdown of $Phf6$ using shRNA during cortex development delayed neuron migration [26], we assessed the possibility of a cortical layering defect. Using SATB2, CTIP2, and TBR1 as markers of cortical layers II-IV, V and VI respectively [43, 44], we stained sections of the E18.5 parietal cortex and enumerated stained cells in 10 pial-ventricular bins. No changes in the distribution or frequency of SATB2, CTIP2, or TBR1 stained cells were observed in E18.5 males with hemizygous germline $Phf6$ mutation (Fig 4) or in females with heterozygous germline $Phf6$ mutation (S8 Fig). Similarly, in the adult brain the distribution of cortical layer markers, reelin, CUX1, CTIP2 and FOXP2 were similar in $Phf6^{lox/Y}$;$Nes$-$cre^{Tg/+}$ compared to control brains (S9 Fig). The foetal and adult cortical layer examination indicated that the germline or nervous system-specific deletion of $Phf6$ does not cause a cortical layering defect.

## No change in neurite number or length after culture for 5 days

To assess whether PHF6 may have an impact on neurite outgrowth and branching, we isolated cortical neurons from E16.5 foetuses and cultured these for 5 days. We found no difference in the number or length of primary, secondary or tertiary neurites (S10 Fig), indicating that loss of PHF6 does not impair the ability of cortical neurons to form neurites *in vitro*.

## Loss of PHf6 causes a reduction in neural stem cell self-renewal and an increase in neuronal differentiation at the expense of astrocyte differentiation

To assess the effects of $Phf6$ deletion on neural precursor cell proliferation, we isolated E12.5 forebrain $Phf6^{lox/Y}$;$Nes$-$cre^{Tg/+}$ and $Phf6^{+/Y}$;$Nes$-$cre^{Tg/+}$ neural stem cells and cultured them as floating colonies (neurospheres) of neural stem and progenitor cells (NSPCs; Fig 5A). Under proliferating conditions with FGF2 and EGF, $Phf6^{lox/Y}$;$Nes$-$cre^{Tg/+}$ NSPCs proliferated normally in culture over 15 passages (S11 Fig). However, $Phf6^{lox/Y}$;$Nes$-$cre^{Tg/+}$ NSPCs plated in serial dilution cultures gave rise to fewer secondary colonies, indicating a reduced capacity to generated colony-forming neural stem cells (Fig 5B). Under differentiating conditions, $Phf6^{lox/Y}$;$Nes$-$cre^{Tg/+}$ and $Phf6^{+/Y}$;$Nes$-$cre^{Tg/+}$ NSPCs formed neurons and astrocytes (Fig 5C). However, $Phf6^{lox/Y}$;$Nes$-$cre^{Tg/+}$ NSPCs formed more neurons (ßIII-tubulin$^+$) and fewer astrocytes (GFAP$^+$, S100ß$^+$) than control cells, assessed as percentage of live cells by flow cytometry (p = 0.02 to 0.002; Fig 5D and S12 Fig), while the percentage of oligodendrocytes (O4$^+$) was similar between genotypes.

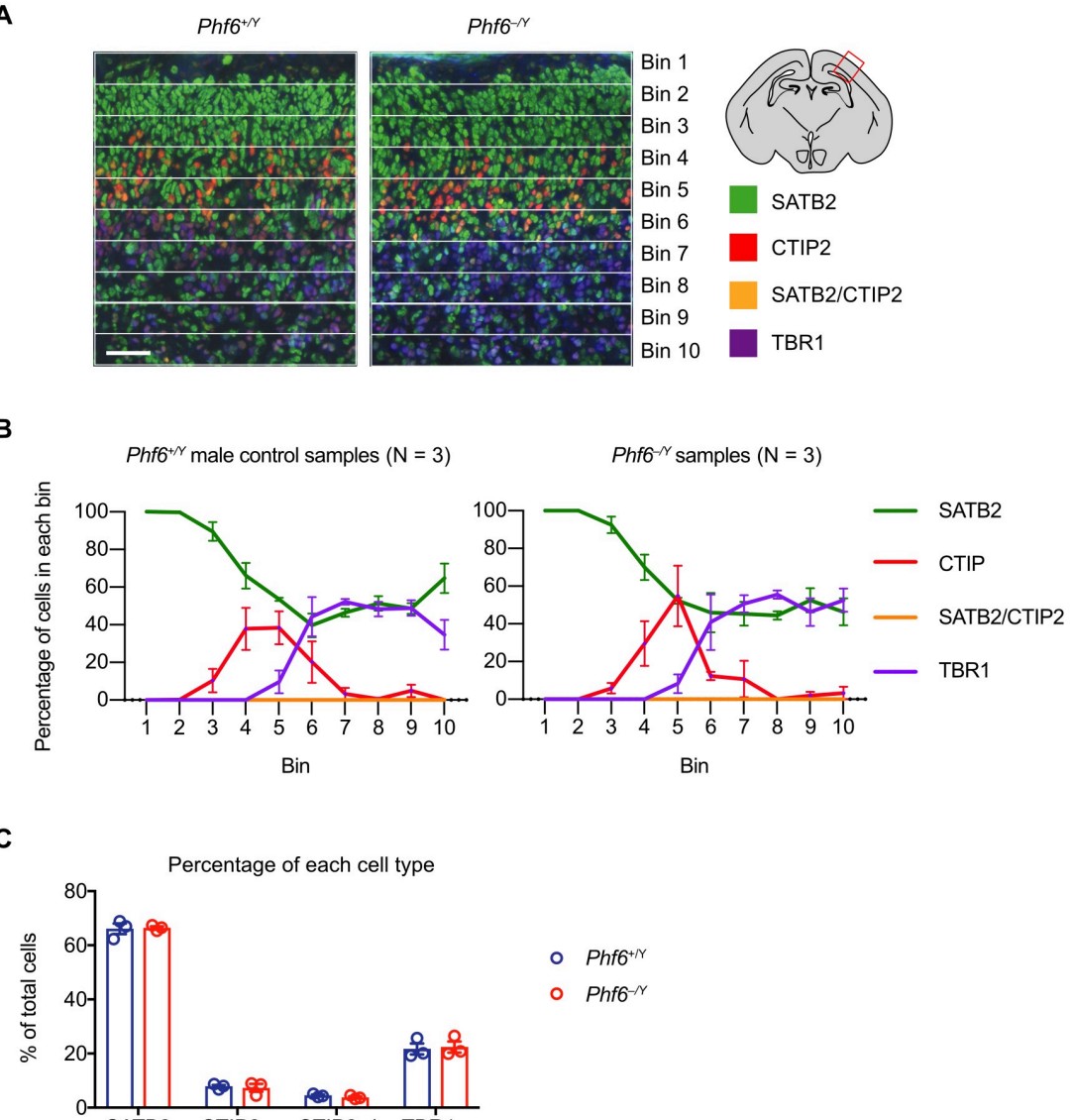

**Fig 4. Germline *Phf6* deletion does not cause a cortical lamination defect.** (A) Representative images of cortical layer staining using SATB2, CTIP2, and TBR1 showing the developing *Phf6⁺/ʸ* and *Phf6⁻/ʸ* E18.5 parietal cortex divided into 10 pial to ventricular bins. Scale bar = 50 µm. (B) Percentage of cells expressing the marker proteins in each bin for three mice per genotype. No significant differences between genotypes were observed. (C) Percentage of cells of each type across the entire thickness of the cortex. No significant differences between genotypes were observed. N = 3 *Phf6⁺/ʸ* and 3 *Phf6⁻/ʸ* foetuses. Data are presented as mean ± sem and were analysed by two-way ANOVA with Šídák's multiple comparisons test. Circles (C) represent individual mouse foetuses. Related data are shown in S8 and S9 Figs; assessment of neurite outgrowths in cultured cortical neurons is displayed in S10 Fig.

## PHF6 regulates gene expression in foetal cortical neurons

PHF6 is a chromatin-associated protein, expected to be involved in transcriptional regulation. Therefore, we examined the effects of PHF6 loss on gene expression in the developing cerebral cortex and in isolated cortical neurons by RNA-sequencing. Total RNA was isolated from *Phf6⁻/ʸ* and control *Phf6⁺/ʸ* E15.5 cerebral cortices, as well as from *Phf6^{lox/Y};Nes-cre^{Tg/+}* and *Phf6^{+/Y};Nes-cre^{Tg/+}* control E16.5 cortical neurons cultured for 24 h to enrich for neuronal cell

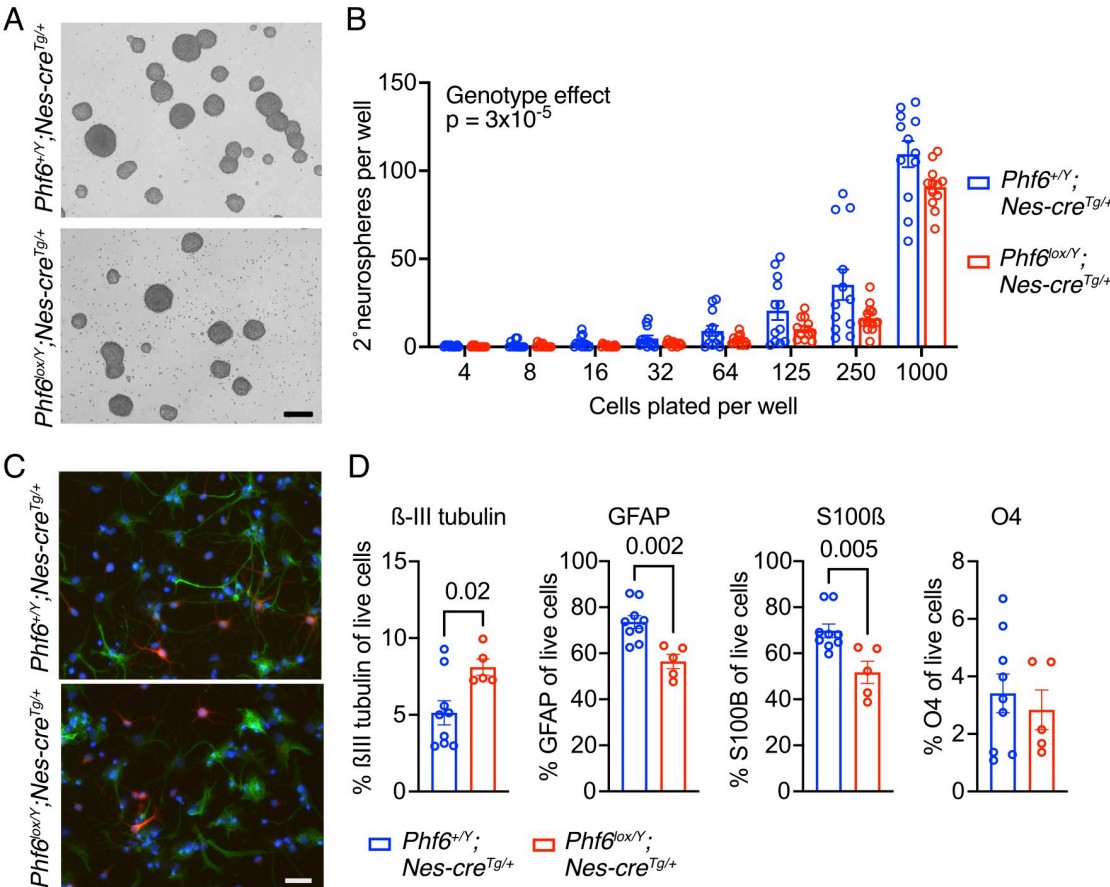

**Fig 5. Neural stem cell self-renewal and differentiation.** (A) Representative bright field images of neural stem and progenitor cells (NSPCs) isolated from the dorsal telencephalon of E12.5 *Phf6⁺/ʸ;Nes-creᵀᵍ/⁺* and *Phf6ˡᵒˣ/ʸ;Nes-creᵀᵍ/⁺* embryos and grown as neurospheres in proliferation medium with FGF2 and EGF. Scale bar equals 220 μm. (B) Secondary neurosphere formation assay of *Phf6⁺/ʸ;Nes-creᵀᵍ/⁺* and *Phf6ˡᵒˣ/ʸ;Nes-creᵀᵍ/⁺* NSPCs plated at the cell numbers per 96-well plate well as indicated. (C) Representative immunofluorescence images of *Phf6⁺/ʸ;Nes-creᵀᵍ/⁺* and *Phf6ˡᵒˣ/ʸ;Nes-creᵀᵍ/⁺* NSPCs plated on laminin coated slides and grown for 5 days in differentiation medium (without FGF2 or EGF and with FCS) stained for a neuronal marker (ßIII-tubulin, red) and an astrocyte marker (GFAP, green) and counterstained with DAPI. Scale bar equals 25 μm. (D) Flow cytometry assessment of the percentage of neuron (ßIII-tubulin⁺), astrocytes (GFAP⁺ and S100ß⁺) and oligodendrocytes (O4⁺) in *Phf6⁺/ʸ;Nes-creᵀᵍ/⁺* and *Phf6ˡᵒˣ/ʸ;Nes-creᵀᵍ/⁺* NSPCs plated on laminin coated slides and grown for 5 days in differentiation medium. N = NSPC cell isolates from 4 embryos per genotype cultured in triplicates (B) and NSPC cell isolates from 9 *Phf6⁺/ʸ;Nes-creᵀᵍ/⁺* and 5 *Phf6ˡᵒˣ/ʸ;Nes-creᵀᵍ/⁺* embryos (D). Data are depicted as means ± sem. Data were analysed by two-factorial ANOVA (B) or unpaired two-tailed Student's t-test (D). Related data are displayed in S11 and S12 Figs.

types. Barcoded libraries were generated and sequenced (S13A and S13B Fig). Using an FDR cut-off of 0.05, we identified 9 downregulated and 51 upregulated genes in the *Phf6⁻/ʸ* vs. *Phf6⁺/ʸ* E15.5 cerebral cortex (Fig 6A and S13C Fig and Tab A in S3 Table) and 10 downregulated and 30 upregulated genes in the *Phf6ˡᵒˣ/ʸ;Nes-creᵀᵍ/⁺* vs. *Phf6⁺/ʸ;Nes-creᵀᵍ/⁺* E16.5 cultured cortical neurons (Fig 6B and S13D Fig and Tab A in S4 Table). There was a positive correlation in the PHF6-regulated genes between the E15.5 and E16.5 datasets (p = 0.0004; Fig 6C). Five genes, *Peg10*, *Phip*, *Zdbf2*, *Nefl* and *Pgap1*, were individually significantly affected in both E15.5 cortex and E16.5 cortical neurons, all five upregulated (Tab A in S3 and S4 Tables). Nine of the top 10 genes differentially expressed in each of the two tissue/cell types were upregulated in the absence of PHF6 (Fig 6D and Tab A in S3 and S4 Tables). Among these genes were *Peg10*, *Phip* and *Zdbf2*. *Reelin* was the top differentially expressed gene, and *Nr4a2* was

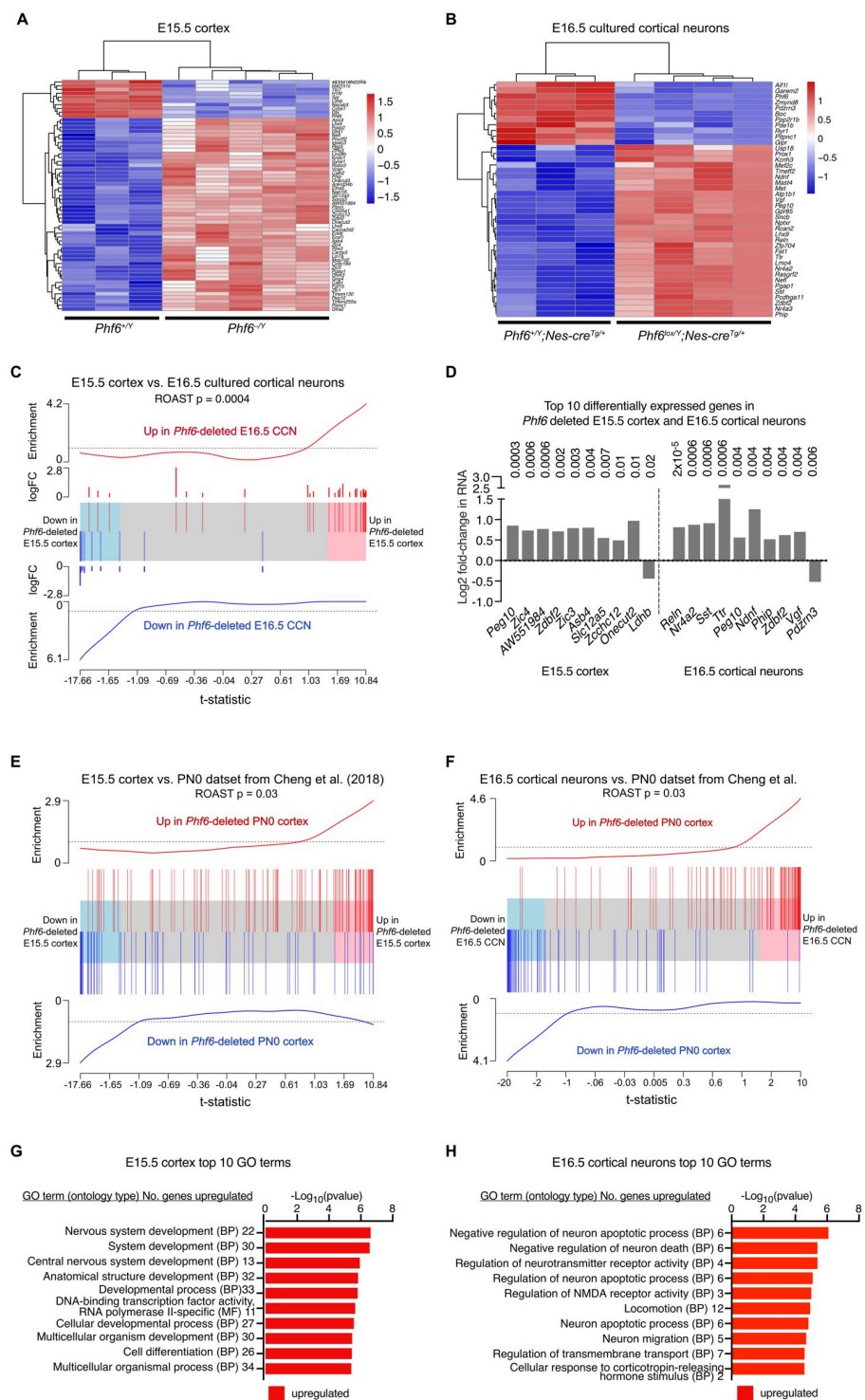

**Fig 6. RNA-sequencing reveals PHF6 regulated genes in the foetal cerebral cortex.** (A) Heatmap showing differentially expressed genes between N = 3 *Phf6*<sup>+/Y</sup> and 5 *Phf6*<sup>-/Y</sup> E15.5 cortices. Values are row-wise z-scores. (B) Heatmap showing differentially expressed genes between N = 3 *Phf6*<sup>lox/Y</sup>;*Nes-cre*<sup>Tg/+</sup> and 4 *Phf6*<sup>+/Y</sup>;*Nes-cre*<sup>Tg/+</sup> E16.5 cortical neuron isolates. Values are row-wise z-scores. (C) Barcode plot showing correlation between the genes differentially expressed in the *Phf6*<sup>-/Y</sup> *vs. Phf6*<sup>+/Y</sup> E15.5 cortex and in the *Phf6*<sup>lox/Y</sup>;*Nes-cre*<sup>Tg/+</sup> vs. *Phf6*<sup>+/Y</sup>;*Nes-cre*<sup>Tg/+</sup> E16.5 cultured cortical neurons. The horizontal axis shows the t-statistic for all genes in the *Phf6*-deleted E15.5 cortex and the blue and red shaded areas are genes that tend towards down- or upregulation respectively in the *Phf6* deleted E15.5 cortex. The vertical lines represent the genes differentially expressed in the E16.5 *Phf6* deleted cultured cortical

neurons. Worms show the relative enrichment of differentially expressed genes in the E16.5 dataset among the differentially expressed genes in the E15.5 dataset. The data sets are positively correlated (genes tend to be similarly up- or downregulated in the same direction). ROAST p-value shown. (D) Log$_2$-fold change in RNA levels of the top 10 differentially expressed genes by FDR in *Phf6*$^{-/Y}$ vs. *Phf6*$^{+/Y}$ E15.5 cortex and *Phf6*$^{lox/Y}$;*Nes-cre*$^{Tg/+}$ vs. *Phf6*$^{+/Y}$;*Nes-cre*$^{Tg/+}$ E16.5 cortical neurons. FDR shown above bars. (E,F) Barcode plots showing correlation between differentially expressed genes for indicated datasets. The horizontal axis shows moderated t-statistics for all genes in either the *Phf6*-deleted E15.5 cortex (E) or the E16.5 cultured cortical neuron dataset (F). The blue and red shaded areas represent genes that tend towards down or upregulation. The vertical lines represent the genes differentially expressed in the *Phf6*-deleted PN0 cortex samples described by Cheng and colleagues [31]. Worms show the relative enrichment of differentially expressed genes of PN0 genes among the differentially expressed genes of the E15.5 and E16.5 datasets. In each case, the data sets are positively correlated, i.e., genes are generally differentially expressed in the same direction. P-values are from ROAST gene set tests. (G,H) Top 10 enriched GO terms in the *Phf6*-deleted E15.5 cerebral cortex (G) and E16.5 cultured cortical neurons (H). BP = biological process, MF = molecular function. The analysis of RNA sequencing data is described in the methods section. FDR < 0.05 was considered significant. Related data are shown in S13 and S14 Figs and in S3 and S4 Tables.

also among the top 10 genes differentially expressed genes in the *Phf6*$^{lox/Y}$;*Nes-cre*$^{Tg/+}$ vs. *Phf6*$^{+/Y}$;*Nes-cre*$^{Tg/+}$ E16.5 cortical neurons. Interestingly, two members of the ZIC family transcription factor genes, *Zic3* and *Zic4*, were among the top 10 genes upregulated in *Phf6*$^{-/Y}$ vs. *Phf6*$^{+/Y}$ E15.5 cortex, and a third member, *Zic1*, was on position 19 (Fig 6D and S13E Fig). One gene associated with enlarged lateral ventricles, *Usp18*, was upregulated in E16.5 cortical neurons lacking PHF6 (S13F Fig). We examined four genes (*Nefl*, *Peg10*, *Phip*, *Zdbf2*) upregulated in E15.5 cortex and E16.5 cortical neurons in the absence of PHF6 in an independent set of samples by RT-qPCR. We found that the increase in mRNA levels by RNA sequencing was replicated in independent *Phf6*$^{lox/Y}$;*Nes-cre*$^{Tg/+}$ vs. *Phf6*$^{+/Y}$;*Nes-cre*$^{Tg/+}$ E15.5 cortices (S14A Fig). The differences in mRNA levels were too subtle to be visible by whole-mount in-situ-hybridisation (S14B Fig). We compared our RNA sequencing datasets with genes reported to be differentially expressed in the *Phf6*-deleted postnatal day 0 (PN0) cortex by Cheng and colleagues [31] and found significant positive correlations for both our datasets (Fig 6E and 6F). Genes tended to be differentially expressed in the same direction, however, a subset of genes upregulated in E15.5 cortex were reported to be downregulated in the PN0 cortex by Cheng and colleagues [31], including *Zic1*, *Zic3*, *Zic4* and *Ecel1*.

To explore cellular and molecular processes affected by loss of PHF6, we performed KEGG and Gene Ontology (GO) analyses to test for terms and pathways enriched among PHF6-regulated genes (Tabs B and C in S3 and S4 Tables). The top 10 significant GO categories enriched for differentially expressed genes in the *Phf6*-deleted E15.5 cortex were enriched among upregulated genes and included the GO terms 'nervous system development' and 'central nervous system development' (Fig 6G). The top 10 significant GO pathways in the E16.5 *Phf6*-deleted cultured cortical neurons were similarly enriched among upregulated genes and included the GO terms 'neuron migration' and 'regulation of neurotransmitter receptor activity' (Fig 6H). These data indicate transcriptional regulation of pathways involving neuronal development and function are affected in the *Phf6* deleted foetal cerebral cortex.

To explore which specific differentially expressed genes were involved in neuronal development or function, we used the Mouse Genome Informatics (MGI) database and additional literature searches to search for neural related phenotypes in mouse models and human disorders involving the nervous system associated with our differentially expressed genes. We identified 25 genes (24 up, 1 down) differentially expressed in the E15.5 cerebral cortex (Tab D in S3 Table) and 18 differentially expressed genes (15 up, 3 down) in the E16.5 cortical neurons (Tab D in S4 Table) that were associated with neural-related phenotypes in mouse models or human disorders, including *Reln*, *Nr4a2*, *Phip*, *Nefl*, *Zic1*, *Zic3*, *Zic4* and *Slc12a5*. This indicates

that PHF6 plays an important role in regulating the mRNA levels of genes important for neural development or function in the foetal cerebral cortex.

## Loss of PHF6 results in precocious calcium signalling in foetal cortical neurons

After observing an increase in neuronal differentiation of $Phf6^{lox/Y}$;$Nes$-$cre^{Tg/+}$ vs. $Phf6^{+/Y}$;$Nes$-$cre^{Tg/+}$ NSPCs, we asked if there could potentially be a functional difference between the two genotypes. We examined spontaneous calcium fluctuations reflecting neuronal activity using calcium imaging and automated analysis of calcium peaks. Cortical neurons were isolated from E16.5 foetuses and cultured for 5 or 7 days in a culture system that preferentially supported the survival of just neurons. Both $Phf6^{lox/Y}$;$Nes$-$cre^{Tg/+}$ and $Phf6^{+/Y}$;$Nes$-$cre^{Tg/+}$ cultures contained a high percentage of excitatory neurons (VGLUT1$^+$; Fig 7A and 7B) and no parvalbumin$^+$ inhibitory neurons. A subset of cells in all cultures displayed calcium signalling peaks (Fig 7C and 7D). Interestingly, despite a moderately reduced percentage of VGLUT1$^+$ cells in $Phf6^{lox/Y}$;$Nes$-$cre^{Tg/+}$ vs. $Phf6^{+/Y}$;$Nes$-$cre^{Tg/+}$ cultures (88% vs. 96%, respectively; p = 0.03; Fig 7B), $Phf6^{lox/Y}$;$Nes$-$cre^{Tg/+}$ contained a larger percentage of neurons displaying minor and major calcium peaks (74% vs. 45% and 44% vs. 19%, respectively; p $< 10^{-6}$ for both; Fig 7E and 7F) after 5 days in culture. After 7 days in culture, the percentages of neurons displaying calcium signalling was similar between the genotypes, suggesting that deletion of $Phf6$ is associated with premature calcium signalling in excitatory neurons. The number of peaks per neuron and the maximum peak height (amplitude) were also increased 1.2-fold in $Phf6^{lox/Y}$;$Nes$-$cre^{Tg/+}$ vs. $Phf6^{+/Y}$;$Nes$-$cre^{Tg/+}$ neurons (p = 0.00002 and 0.02, respectively; Fig 7G and 7H). While the height of minor peaks was modestly increased in $Phf6^{lox/Y}$;$Nes$-$cre^{Tg/+}$ vs. $Phf6^{+/Y}$;$Nes$-$cre^{Tg/+}$ neurons by 8% (p $< 10^{-6}$; Fig 7I), the height of major peaks was modestly decreased in $Phf6^{lox/Y}$;$Nes$-$cre^{Tg/+}$ vs. $Phf6^{+/Y}$;$Nes$-$cre^{Tg/+}$ neurons by 9% (p $< 10^{-6}$; Fig 7J). Lastly, the number of minor peaks per neuron was increased in $Phf6^{lox/Y}$;$Nes$-$cre^{Tg/+}$ vs. $Phf6^{+/Y}$;$Nes$-$cre^{Tg/+}$ cortical neuron cultures per foetus, while the number of major peaks per foetus was not different between genotypes (S15 Fig).

Overall, our calcium imaging data suggest that $Phf6^{lox/Y}$;$Nes$-$cre^{Tg/+}$ cortical neurons precociously commence calcium signalling with a higher percentage of cells signalling earlier, more peaks per neuron, in particular more lower amplitude peaks per neuron.

## Discussion

In this paper, we report the characterization of neural-associated phenotypes in an animal model of male and female BFLS. We show that spontaneous seizures occur in mice with either germline or nervous system-specific deletion of $Phf6$, demonstrating that seizures occur via a neural intrinsic mechanism. In addition, we show that cortical neurons lacking PHF6 spontaneously undergo premature calcium signalling. Previously, Cheng and colleagues showed that stellate neurons in fresh brain slices isolated from mice with the $Phf6^{C99F}$ hypomorphic mutation displayed hyper-excitability in response to automatic current injection [31]. Thus, hyper-excitability may underpin the neural intrinsic seizure susceptibility resulting from $Phf6$ mutations. Among 20 individuals with BFLS, four presented with seizures, including a female with exon 4 and 5 deletion [9] (the same exons deleted in our mice), two males, both with p. E338del, and one female with p.G248V.

Sites of mildly abnormal lamination and heterotopic neurons have been reported in one male BFLS patient [29], while two female BFLS patients displayed possible subcortical band heterotopia [30]. It is unclear if such cortical abnormalities are common in individuals affected by BFLS, due to the paucity of imaging studies. Our histological and immunofluorescence

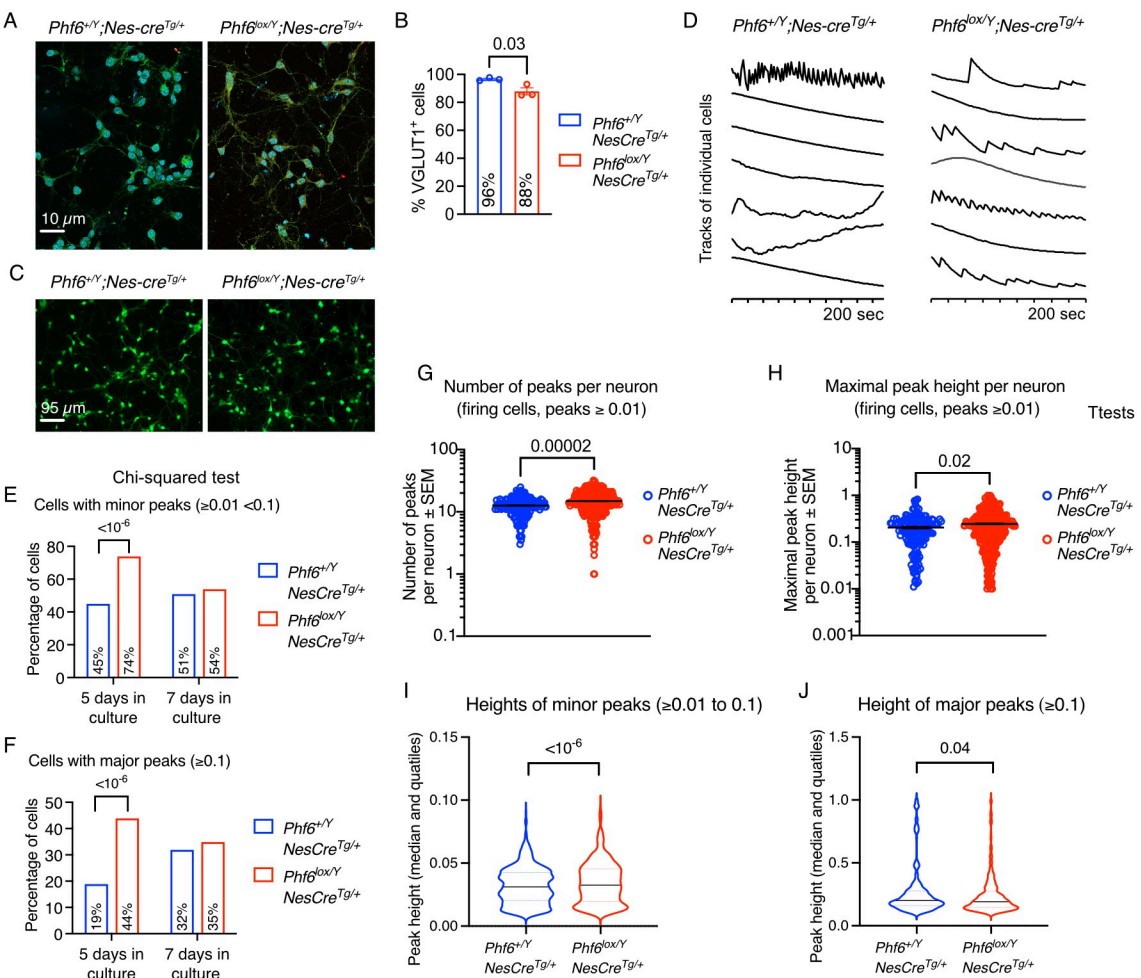

**Fig 7. *Phf6* deletion causes calcium fluctuations prematurely indicative of precocious neuronal activity in foetal cortical neurons.**
(A) Representative immunofluorescence images of cortical neurons isolated from E16.5 *Phf6^{lox/Y};Nes-cre^{Tg/+}* and *Phf6^{+/Y};Nes-cre^{Tg/+}* foetuses detecting the excitatory neuron marker vesicular glutamate transporter 1 (VGLUT1; green) counterstained with DAPI (blue) after 5 days in culture. (B) Percentage of *Phf6^{lox/Y};Nes-cre^{Tg/+}* and *Phf6^{+/Y};Nes-cre^{Tg/+}* cells positive for VGLUT1 staining in cultures of cortical neurons. (C) Representative fluorescence images of cortical neurons isolated from E16.5 *Phf6^{lox/Y};Nes-cre^{Tg/+}* and *Phf6^{+/Y};Nes-cre^{Tg/+}* foetuses loaded with a Ca^{2+} responsive dye (Fluo-4, AM) to reveal high intracellular calcium concentrations (green). (D) Representative calcium imaging tracks of individual neurons isolated from E16.5 *Phf6^{lox/Y};Nes-cre^{Tg/+}* and *Phf6^{+/Y};Nes-cre^{Tg/+}* cortices assessed after 7 days of culture in vitro. (E,F) Percentage of *Phf6^{lox/Y};Nes-cre^{Tg/+}* and *Phf6^{+/Y};Nes-cre^{Tg/+}* cortical neurons displaying minor (1% to <10% of the maximal amplitude; E) or major calcium signalling peaks (≥10% of the maximal amplitude; F) assessed after 5 and 7 days of culture in vitro. (G) Number of calcium signalling peaks per neuron in neurons displaying peaks above background (≥1% of maximal amplitude) in *Phf6^{lox/Y};Nes-cre^{Tg/+}* and *Phf6^{+/Y};Nes-cre^{Tg/+}* cortical neuron cultures assessed after 5 to 7 days in vitro. (H) Maximum peak height per neuron in neurons displaying peaks above background (≥1% of maximal amplitude) in *Phf6^{lox/Y};Nes-cre^{Tg/+}* and *Phf6^{+/Y};Nes-cre^{Tg/+}* cortical neuron cultures assessed after 5 to 7 days in vitro. (I,J) Height of minor (1% to <10% of the maximal amplitude; I) and major peaks (≥10% of the maximal amplitude; J) in *Phf6^{lox/Y};Nes-cre^{Tg/+}* and *Phf6^{+/Y};Nes-cre^{Tg/+}* cortical neuron cultures assessed after 5 to 7 days in vitro. N = cortical neurons isolated from 3 *Phf6^{+/Y};Nes-cre^{Tg/+}* and 3 *Phf6^{lox/Y};Nes-cre^{Tg/+}* foetuses (A,B) and 906 *Phf6^{+/Y};Nes-cre^{Tg/+}* and 932 *Phf6^{lox/Y};Nes-cre^{Tg/+}* cortical neurons from 8 *Phf6^{+/Y};Nes-cre^{Tg/+}* and 4 *Phf6^{lox/Y};Nes-cre^{Tg/+}* foetuses (C-J). Data are displayed as mean ± sem (B,G,H), percentage of cells (E,F) or median and quartiles (I,J). Circles represent individual foetuses (B) or individual neurons (G,H). Data were analysed by unpaired Student's t-test (B,G,H), Chi-squared test (E,F) and two-way ANOVA with cell and genotype as the independent factors (I,J). Related data are displayed in S15 Fig.

marker analysis revealed an enlargement of the lateral ventricles in mice lacking PHF6 but indicated that cortical lamination is largely unaffected, in agreement with two other studies of mouse models of BFLS [32,33]. In contrast, a previous report using an shRNA to induce *Phf6* knock down in mice *in utero* resulted in a cortical neuron migration defect and development

of white matter heterotopias. Similarly, a miRNA (miR-128) that targets *Phf6* results in delayed neuron migration [45]. Similar discrepancies between knock-out and shRNA-mediated knock-down cortical layering phenotypes have previously been observed for genes including *Dcx* [46–48] *Crpm2* [49] and *Foxp2* [50,51]. There are several possible explanations for a more severe phenotype being present after shRNA knockdown vs. genetic deletion. One possibility is there might be subtle cortical layering anomalies that were not evident in our experiments but could possibly be detected by thymidine analogue birth dating experiments. Another possibility is that there might be off-target effects of the shRNA, either through artefacts caused by unintended targeting of other genes or through disruption of the endogenous miRNA network [48], which has been shown to explain the severe migration defect caused by shRNA knock-down of *Dcx* [47] that is not present in the complete knockout [46]. An alternative explanation for the discrepancy observed here is that there may be genetic compensation. Recently, it has been demonstrated that nonsense mediated decay (NMD) can trigger the transcriptional activation of genes similar to the mutant mRNA [52–54]. This compensatory mechanism resulted in less severe phenotypes in mutants with NMD but does not occur with mutations that do not cause NMD, nor after genetic knockdown using morpholino constructs [52–54]. Thus, it is possible that such a compensatory mechanism triggered by NMD in our *Phf6* mouse model could explain the less severe phenotype compared with *Phf6* knockdown. Notwithstanding, a number of BFLS mutations are truncating [6] and would similarly be expected to result in NMD, rendering our mouse model clinically relevant for at least a subset of patients. Supporting this we have shown reduced PHF6 protein levels in cells derived from individuals with BFLS, including in those with protein truncating mutations.

Two female BFLS patients reported to have aspects resembling subcortical band heterotopia had duplications of exon 4 and 5 [30], which is predicted to result in a frameshift mutation [4]. In three female BFLS individuals, a whole gene deletion was associated with ventricular dilation and increased periventricular white matter, p.C99* with hypoplasia of the corpus callosum and cavum septum pellucidum and p.G248V with cortical dysplasia involving blurring of grey-white matter junction in the right frontal lobe; two males (p.E338del; p.H229R) had reduced or delayed myelination [9]. MRI examination of two female BFLS individuals, one with a whole gene deletion and the other with an exon 9 and 10 deletion, revealed wide ventricles [4,7]. A male BFLS individual with reported areas of indistinct lamination [29] was from the first described BFLS family and therefore would have harboured a p.R342* truncating mutation at the C-terminus of the protein [36]. We have shown here that this mutation results in very low levels of truncated PHF6 protein. The abnormalities described by Brun and colleagues [29] were noted by Kasper and co-workers to be milder than in the two female BFLS patients that they described, mentioned above [30]. Kasper and colleagues did not observe abnormalities in MRI datasets from two other female BFLS patients; one with mutation p.G226fs*53 [4,55] and the other with a 100 kb deletion covering the last 5 coding exons of *PHF6* [56]. Similarly, no structural brain anomalies in four of nine BFLS individuals with mutations p.L244F, p.R319*, p.E338del and p.R342* examined by MRI [9]. It was previously noted that there did not appear to be a clear genotype-phenotype correlation in BFLS [6]. Studies in further BFLS patients or additional animal models with patient specific mutations will be required to clarify how widespread cortical abnormalities and heterotopias are after *PHF6* mutation and may enable conclusions to be drawn about a possible genotype/phenotype correlation.

We observed that *Phf6* deleted neural stem cells were less capable of self-renewal. In contrast, *Phf6* deleted neural stem cells have been reported to undergo increased self-renewal [33]. This discrepancy could relate to differences in the type on NSPCs examined in the two studies. Here, we examined embryonic E12.5 dorsal telencephalon neural stem cells, whereas Rasool

and colleagues used tamoxifen induced Cre-ERT2 deletion of *Phf6* at E14.5 and isolated neural stem cells from the E15.5 to E16.5 foetal dorsal and ventral forebrain and midbrain. The two sources of cells are different with respect to developmental stage and anatomical location. Different embryonic and foetal stages [57] and brain locations [58] give rise to distinct populations of neuronal stem cells. E12.5 forebrain neural precursors, although not a major source of adult neural stem cells, contribute to neuronal populations of the cortex, hippocampus, septum, striatum and olfactory bulbs, whereas subsets of E15.5 forebrain neural stem cells are unipotent, giving rise to glia or neurons, while other subsets give rise to neurons and glia [57]. Neural stem cells retain expression of the developmental patterning genes of their anatomical location into adulthood, even while able to give rise to diverse progeny in distant locations [59]. Lastly, in some studies, neural stem cells were restricted to cell fates from their location of origin, whereas in other studies greater plasticity was observed [59,60]. Overall, these studies indicate that the source and developmental stage of the neural stem cells could account for the differences observed between the results in this study and Rasool and colleagues' work [33]. Alternatively, differences in experimental conditions may have caused the discrepancies in results.

Our observation of a reduced self-renewal capacity was accompanied by an increase in neuronal differentiation and paralleled by precocious spontaneous calcium signalling in isolated E16.5 cortical neurons, suggesting premature maturation of neurons, which might explain the spontaneous epileptic seizures that we observed in *Phf6* deleted mice, which mirrored spontaneous seizures occurring in patients, that were not reported in other *Phf6* mouse models. In addition, reduced neural progenitor self-renewal and premature neuronal differentiation could deplete the progenitor pool and thereby contribute together with other factors to the reduction in cerebral cortex size observed in adult mice.

Our RNA-sequencing analysis of E15.5 cortex and E16.5 cortical neurons has identified a number of genes differentially expressed upon PHF6 loss. While it remains to be established whether the differentially expressed genes are direct PHF6 targets, the greater number of upregulated genes compared to downregulated genes in both E15.5 and E16.5 *Phf6*-deleted samples leads us to speculate that PHF6 functions as a transcriptional repressor. A role for PHF6 in transcriptional repression is consistent with the association of PHF6 with the nucleosome remodelling and deacetylase complex (NuRD), which generally acts as a transcriptional repressor [61]. However, the CHD4 chromatin remodelling subunit of NuRD is also involved in gene activation [62], and PHF6 has also been associated with the polymerase associated factor 1 (PAF1) transcriptional elongation complex in cortical neurons [26], which suggests that PHF6 may also play a role in the positive regulation of gene expression in neurons. Accordingly, PHF6 was observed to be required for the expression of ephrin receptor genes in neural stem cells [33]. Additional studies are needed to elucidate the molecular effects of PHF6 loss on gene transcription. PHF6 has been shown to have context-specific roles in tumour suppression [12,63] and thus it is likely PHF6 may also play stage and context-specific roles in brain development.

The individually significant upregulation of five genes due to loss of PHF6 overlapped between the E15.5 cortex and the E16.5 cortical neurons, including *Nefl* and *Phip*. Increased expression of the neurofilament gene, *Nefl*, in mice leads to neurofilament accumulation, axonal swelling and degeneration [64]. *Phip* (pleckstrin homology domain protein 1) mRNA was moderately upregulated in both the E15.5 cerebral cortex and E16.5 cortical neurons (as well as the PN0 cortex). Its protein product has been shown to interact with PHF6 by co-immunoprecipitation [65]. The PHIP protein contains a tudor domain and binds mono- or tri-methylated histone 3 lysine 4 [65]. It is possible that loss of PHF6 may trigger a feedback mechanism that increases expression of *Phip*. While loss of function *PHIP* mutations have been identified

in human intellectual disability [66,67] and murine *Phip* loss of function mutations cause post-natal growth retardation [68], the effects of upregulation of *Phip* remain to be investigated.

Among the top 10 genes regulated by PHF6 in each tissue/cell type, were genes known to be involved in brain development and function. Apart from *Phip*, already discussed above, these included *Slc12a5* and ZIC family transcription factor genes in E15.5 cortex, as well as *reelin* and *Nr4a2* in E16.5 cortical neurons. It is important to note, that these genes were upregulated in the absence of PHF6, whereas their well-known brain development phenotypes are often associated with a partial or complete loss of function, and the effects of gain of function have not been reported in all cases. Notably, the increases in mRNA level observed here in the *Phf6* mutant tissue and cells are subtle compared to overexpression levels commonly achieved in transgenic mice, and therefore, one would not expect in the *Phf6* deleted brains some of the drastic effects observed in transgenic overexpressing mice. In the *Phf6* null E15.5 cortex, upregulation of three members of the ZIC family of transcription factor genes stood out. Complete loss of *Zic1* [69] or *Zic3* [70] causes cerebellum defects. *Zic1* and *Zic3* double deleted mice also display forebrain defects [71]. Loss of *Zic4* causes cerebellar hypoplasia [72] and combined haploinsufficiency of the linked *ZIC1* and *ZIC4* genes causes Dandy-Walker cerebellar malformation in humans and cerebellar defects in mice [73]. Gain of function of *ZIC1* is associated with craniosynostosis and learning disability [74]. The effects of overexpression of the other two ZIC genes during development remain to be investigated. SLC12A5A/KCC2 is required for synaptic inhibition [75] and loss of *SLC12A5* is associated with epilepsy in a humans [76] and in mice [77]. The effects of overexpression of *Slc112a5* during brain development remain to be examined. *Reelin* mRNA was increased in *Phf6* deleted cortical neurons. Interestingly, while loss of *reelin* is associated with lissencephaly in humans [78] and neuron migration and cortical cerebellar layering defects in mice [79], overexpression of *reelin* also causes cortical neuron migration and cortical layering defects [80]. Mice lacking the orphan nuclear receptor NR4A2/NURR1 fail to develop midbrain dopaminergic neurons [81], while the effects of gain of NR4A2 during brain development have not been reported. While the overexpression studies of most of the genes upregulated due to loss of PHF6 has not been reported, gain of function commonly affects the same organ system as loss of function and therefore, disturbance of brain development due to upregulation of these genes in the absence of PHF6 is likely. The combined effect of upregulation of a number of these genes may contribute to the intellectual disability in individuals affected by BFLS. The datasets reported in this paper combined with the gene expression of the postnatal cerebral cortex reported by Cheng and colleagues [31], as well as Ahmed and colleagues [32] form a developmental time-course of genes transcriptionally affected by *Phf6* deletion, providing a resource for further investigation into the role of PHF6 in cerebral cortex development.

In conclusion, we report the first observations of spontaneous seizures, increased neuronal differentiation and increased spontaneous neuronal activity in an animal model of BFLS. Furthermore, we report a number of disrupted pathways and differentially expressed genes involved in neural function and/or development in the absence of PHF6. Our data identify cell biological and molecular defects in mice lacking PHF6 that may also be affected in BFLS patients.

## Methods

### Ethics statement

All work on human subjects was approved by the Women's and Children's Health Network Human Research Ethics Committee and informed consent was obtained from all participants.

All animal experiments were performed according to the Australian code of practice for the care and use of animals for scientific purposes with the approval of the Walter and Eliza Hall Institute Animal Ethics Committee.

## Human samples

Epstein-Barr virus (EBV) immortalized B-cell lines (LCL) were established from peripheral blood lymphocytes of patients as previously described [82]. Once established, LCL were cultured in RPMI 1640 (Invitrogen, Australia) supplemented with 10% FCS, 2 mM L-glutamine, 0.017 mg/mL benzylpenicillin and grown at 37˚C with 5% $CO_2$.

## Animals

The *Phf6* targeted allele is as previously described with *loxP* sites flanking exons 4 and 5 of the *Phf6* gene, which results in undetectable levels of PHF6 in tissues, which have undergone cre-mediated recombination [12]. Cre-deleter mice [83] were used for germline deletion of *Phf6*. *Nestin-cre* transgenic mice [38] were used for nervous system specific deletion, which mediates recombination in the neural tube from E11.5 [39]. Genotyping primers for *Phf6* and *Cre* have been reported previously [12]. All mice in this study were on the C57BL/6 background. The incidence and details of seizures, hydrocephalus and other anomalies were monitored and recorded. The analyses of hydrocephalus incidence included all animals of the relevant genotypes in each colony while the seizure incidence included all animals kept over 100 days.

The mice were observed daily at specific times of the day. Mice displayed unprovoked, spontaneous seizures (no light, no sound, no other form of interference), but also seizures in response to being disturbed, for example by the cage cleaning process. In general, mice with a tendency to suffer spontaneous seizures will display seizures specifically when disturbed, e.g., during cage cleaning. Since the cage cleaning took place 2x/week, we have determined the onset of seizures within ~3.5 days accuracy. Since the latency is quite long (median latency of ~600 days), this corresponds to an accuracy of approximately ± 0.6%. The seizures lasted between 2 and 10 minutes and recurred with varying periodicity.

## Western blotting of human and mouse samples

Cells were lysed by incubation in radioimmunoprecipitation assay (RIPA) buffer (65.3 mM Tris [pH 7.4], 150 mM NaCl, 1% Nonidet P40, 50 mM NaF, 0.1 mM $Na_3VO_4$ and 1 x protease inhibitor cocktail [Sigma]) on ice for 15 minutes followed by trituration through a 21G needle. Insoluble material was removed by centrifugation at 13, 000 x *g* for 15 minutes. Cleared lysates were separated on 4–12% Bis-Tris polyacrylamide gel (Invitrogen) then transferred onto Bio-Trace NT nitrocellulose membranes (PALL Life Sciences). Membranes were blocked with 5% skim milk, washed, then incubated with anti-PHF6 antibody clone 4B1B6 (1.5 mg/ml) [17] and subsequently with goat anti-mouse IgG HRP-conjugated antibody (Dako). ß-tubulin (AbCam ab6046) and Ponceau S staining were used as loading control. Enhanced chemilumi-nescence method (GE Healthcare, Sweden) was used to develop the blots. Examination of the blots showed that ß-tubulin was present in variable amounts in the lymphoblastoid cell lines derived from different individuals. Therefore, Ponceaus S staining of the gel is displayed as a loading control.

Protein lysates from E15.5 whole mouse brain were isolated using KALB lysis buffer (1% Triton X, 1 mM EDTA, 150 mM NaCl, 50 mM Tris HCl pH 7.4, 0.02% $NaN_3$) with added pro-teinase inhibitors (Roche). Denatured lysates were separated by electrophoresis on a 4–12% NuPAGE Bis-Tris gel and transferred onto a PVDF membrane (Osmonics). The membrane was blocked with 5% skim-milk/PBST, probed with rabbit anti-PHF6 (Bethyl Laboratories,

A301-452A), mouse anti-α Tubulin (Invitrogen, A11126) and subsequently HRP conjugated anti-rabbit IgG (Sigma, NA934) and HRP conjugated anti-mouse IgG (Sigma, NA931). Enhanced chemiluminescence method (GE Healthcare, Sweden) was used to develop the blots. The signal was detected using ChemiDoc (BioRad).

## Cresyl-violet staining

Adult mouse brains were perfusion fixed with paraformaldehyde, paraffin-embedded, cut into 10 μm serial sections followed by cresyl-violet staining. Calculation of the lateral ventricle, cortex and overall brain volume was performed using methods adapted from previous studies [84]: ImageJ was used to trace the area of the lateral ventricles, cortex area or total brain area across seven sections for each animal spaced approximately 480 μm apart. Note that because of the different size of the brains between genotypes, representative sections were matched for measurement rather than being randomly selected to ensure the same number of sections were analysed in spanning the region of interest. The volume was calculated by multiplying the sum of the areas ($mm^2$) by 0.01 mm (section thickness) by the number of sections in between each analysed section. Purkinje cells were quantified using ImageJ by counting the number of Purkinje cells per mm of Purkinje cell layer across four evenly spaced sections of the cerebellum for each animal.

## GAD67 staining

GAD67 staining was performed using formaldehyde perfusion-fixed brains as previously described [85] using the Chemicon anti-GAD67 antibody (available from Millipore Cat. #MAB5406, Lot #2775517). Quantification was performed by counting the number of positively stained cells in the cerebral cortex of representative parietal sections of the brain.

## Cortical layer analysis

Foetal cortical layering staining was performed on cryosections of E18.5 brain frozen in OCT Compound (Agar Scientific) as described previously [86]. In brief, following fixation in 4% formaldehyde for 20 minutes, permeabilisation in 0.25% Triton X-100 for 20 min, sections were blocked with 10% normal goat serum for 30 min then stained with a combination of anti-SATB2 (Abcam, ab51502), anti-CTIP2 (Abcam, ab18465), anti-TBR1 rabbit antibody (Abcam, ab31940). After 1 h of primary antibody incubation, slides were washed, then stained with a mix of anti-rabbit IgG AMCA (Jackson ImmunoResearch, 111-156-003), anti-rat IgG Alexa 568 (Invitrogen, A-11077), and anti-mouse IgG Alexa 488 (Invitrogen, A-11029). Slides were coverslipped with Dako fluorescence mounting medium (Agilent) and imaged using a compound fluorescence microscope (Axioplan2, Carl Zeiss) and a digital camera (AxioCam HRC, Carl Zeiss) utilizing the AxioVision software (v 3.1, Carl Zeiss). Quantification was performed by dividing the span of the cortex into 10 bins and counting the number of cells of each type within each bin.

Adult cortical layering staining was performed on paraffin sections. Briefly, deparaffinised sections were processed through intervening PBS washes with $H_2O_2$ to block endogenous peroxidase, with primary antibody (against Reelin, G20, ab78540, Abcam; CTIP2, 25B6, ab18465, Abcam; CUX1/CASP, 2A10, ab54583, Abcam; FOXP2, ab16046, Abcam), biotinylated secondary antibody (anti-mouse/rabbit IgG, Vector Laboratories), avidin:biotinylated horseradish peroxidase complex (Vector Laboratories) and diaminobenzidine.

## Cortical neuron cultures

Dissected E16.5 cerebral cortices were incubated on ice in pancreatin trypsin (25 mg/ml pancreatin, Sigma, and 5 mg/ml trypsin; Gibco, in PBS) for 30 min, followed by a 5 min incubation at 37°C after removal of the pancreatin trypsin. Alternatively, cortices were incubated with trypsin-EDTA (Sigma-Aldrich, 59430C) and for 10 to 15 min in a 37°C water bath. After rinsing, cortices were dissociated in 2% FCS using a P1000 pipette, then filtered through a 40 μm sieve in a 0.2% BSA/PBS solution. Cells were centrifuged twice for 5 min at $140 \times g$, rinsing with 0.2% BSA/PBS in between, and resuspended in medium (DMEM, 5 mM HEPES, 13.4 mM NAHC0$_3$, 0.6% D-glucose, 25 μg/mL insulin, 100 μg/mL apotransferrin, 9.6 μg/mL putrescine HCL (9.6 μg/mL, 2 mg/mL BSA, 1% FCS, 5.18 nM selenium and 6.3 ng/mL progesterone, 100 μg/mL streptomycin and 100 U/mL penicillin) and filtered again through a 40 μM sieve onto plates pre-coated with poly-L-lysine (Sigma). Alternatively, 50,000 cells/cm$^2$ were plated on 8-well chamber slides (Ibidi, 80826) or 10,000 cells/cm$^2$ on 13 mm coverslips, both pre-coated with 15 μg/ml poly-L-ornithine (Sigma-Aldrich, P4957) for 2 h at 37°C and 5% CO$^2$, followed by 4.5 μg/ml laminin (Sigma-Aldrich, L2020) O/N at 37°C and 5% CO$^2$, with a PBS wash in between coating solutions.

## Immunofluorescence staining and neurite quantification

Cells were cultured as above on poly-L-lysine-coated chamber slides for 5 days. Cells were fixed with 4% formaldehyde for 20 min, permeabilised with 0.3% Triton X-100, blocked in 10% foetal calf serum for 1h, then incubated with anti-β tubulin III (Promega, G7121) in 10% FCS for 1 h at 37°C, followed by anti-mouse IgG Alexa 568 (Molecular probes, A21124) and DAPI in 10% FCS at 37°C for 30 min. Slides were mounted in Dako fluorescence mounting medium and imaged using a compound fluorescence microscope (Axioplan2, Carl Zeiss). Neurite length was traced in Adobe Illustrator. 100–150 neurons were traced per animal.

Alternatively, cells were stained using mouse anti-parvalbumin (Sigma-Aldrich, P3088) and rabbit anti-VGLUT1 (Invitrogen, 48–2400) in 10% (w/v) FBS and 0.25% (w/vol) Triton-X 100 O/N at 4°C. Slides were washed in PBS (2x 5 min) and incubated with goat anti-mouse IgG1 (Alexa Fluor 568, Invitrogen, A21124) and goat anti-rabbit IgG (H+L, Alexa Fluor 488, Invitrogen, A11008) secondary antibodies in 10% (w/vol) NGS for 1 h at 37°C.

## Neural stem and progenitor cell culture

Primary neural stem and progenitor cells (NSPCs) were dissected from the SVZ of adult mice or dorsal telencephalon of E12.5 embryos. Tissue was triturated into a single cell suspension in 1 ml NeuroCult Proliferation Kit (Mouse & Rat; Stemcell Technologies, 05702) with added EGF and bFGF and passed through a 100 μm cell sieve (Corning, 431751). Cells were plated on uncoated 6-well cell culture plates (Corning, 353046) containing 2 ml complete neurosphere medium and grown at 37°C and 5% CO$^2$. Cells were passaged every 4 to 5 days. Floating neurospheres were collected and centrifuged at 200 g for 5 min. Supernatant was removed and cells were resuspended in 1 ml Accutase (StemCell Technologies, 07922) and incubated for 5 min in a 37°C water bath. Cells were resuspended in 5 ml complete neurosphere medium and centrifuged at 200 g for 5 min. Supernatant was removed and cells resuspended in 1 ml fresh complete neurosphere medium and counted using a Countess automated cell counter (Invitrogen), excluding dead cells using trypan blue (Sigma, T8154). Cells were seeded at 50,000 cells in 2 ml fresh complete neurosphere medium. The proliferative capacity of neurosphere cultures was assessed using cell counts at each passage.

## Neurosphere forming assay

Secondary neurosphere formation was assessed as described [87]. Briefly, primary neurosphere cultures were dissociated, cell counts determined, plated into 96-well plates in 100 μl complete neurosphere medium at densities of 4, 8, 16, 32, 64, 125 and 1000 cells per well, in triplicate. Cells were grown in 37˚C and 5% $CO_2$. Secondary neurospheres were counted after 7 days of culture.

## Neural stem and progenitor cell differentiation

Primary neural stem cells were dissected from the dorsal telencephalon of E14.5 foetuses. Tissue was triturated into a single cell suspension in 1 ml complete neurobasal differentiating medium (Gibco) and passed through a 100 μm cell sieve (Corning, 431751). To determine the differentiation capacity of neurospheres, 10,000 cells/cm$^2$ were plated onto 4-well chamber slides (Thermo Scientific, 154526) pre-coated with 15 μg/ml poly-L-ornithine (Sigma-Aldrich, P4957) for 2 h at 37˚C and 5% $CO^2$, followed by 4.5 μg/ml laminin (Sigma-Aldrich, L2020) O/N at 4˚C, with a PBS wash in between coating solutions. Cells were grown in complete neurobasal differentiating medium for 5–7 days at 37˚C and 5% $CO^2$.

## Immunocytochemistry

Cells were cultured in 4-well chamber slides (Thermo Scientific, 154526) pre-coated with 15 μg/ml poly-L-ornithine (Sigma-Aldrich, P4957) for 2 h at 37˚C and 5% $CO^2$, followed by 5 μg/ml laminin (Sigma-Aldrich, L2020) O/N at 4˚C, with a PBS wash in between coating solutions. Cells were cultured under differentiation conditions. Differentiated cells were fixed in 4% (w/vol) PFA for 20 min at RT, washed in PBS (2x 5 min) and blocked in 10% (w/vol) FBS at 37˚C for 1 h. Cells were incubated with mouse anti-βIII tubulin IgG1 (Promega, G7121) and rabbit anti-GFAP (Agilent Technologies, Z033401-2) in 10% (w/v) FBS and 0.25% (w/vol) Triton-X 100 O/N at 4˚C. Slides were washed in PBS (2x 5 min) and incubated with goat anti-mouse IgG1 (Invitrogen, A21124) and goat anti-rabbit IgG (H+L, Alexa Fluor 488, Invitrogen, A11008) secondary antibodies in 10% (w/vol) NGS for 1 h at 37˚C. Slides were washed in PBS (2x 5 min) and the walls of chamber slides removed. Slides were mounted in mounting medium (VECTASHIELD Vibrance, Vector Laboratories, H17002) and allowed to dry O/N at 4˚C. Slides were imaged using a fluorescent microscope (Axioplan2, Zeiss) and a digital camera (AxioCam HRC, Zeiss).

## Flow cytometry

Cells were gently pipetted off the bottom of a 6-well plate and collected by centrifugation (200 g, 5 min) and washed in PBS. Cells were stained with Live/Dead fixable violet dye (1:1000, Thermofisher, L34955) for 30 minutes on ice. Cells were washed in PBS and collected by centrifugation (200 g, 5 min). Cells were fixed and permeabilised using Foxp3 kit fixation/permeabilisation kit (Ton Biosciences, TNB-1022-L160) for 1 h on ice. Cells were washed in 2% FACS buffer (2x 30 min washes at RT). Cells were stained with antibodies detecting O4 (A594, R&D, FAB1326T), βIII-tubulin (PerCP Cy5.5, R&D, IC1195C), SOX2 (APC, R&D, IC2018A), GFAP (A488, Invitrogen, 53-9892-82) and S100β (1:400, Millipore, Ab52642) in 2% FACS buffer O/N at 4˚C. Cells were washed in 2% FACS buffer (2x 15min), followed by wash with 2% FACS buffer O/N at 4˚C. Cells were collected by centrifugation (200 g, 5 min) and incubated with goat anti-rabbit antibody (Alexa Fluor A555, ThermoFisher, A21428) in 2% FACS buffer for 2 h at RT. Cells were washed in 2% FACS buffer (2x 30 min) and resuspended in 200 μl 2%

FACS buffer and analysed on the BD LSRFortessa X-20 Cell Analyzer at fewer than 7500 events/sec.

## RNA isolation

RNA was extracted from dissected E15.5 mouse cortex and E16.5 cortical neurons cultured for 24 hours using a Rneasy Mini Kit (Qiagen, 74104) as per the manufacturer's instructions with the optional Dnase digest step included. RNA quality and concentration were determined by automated electrophoresis using the Agilent 4200 Tapestation (Agilent).

## RNA-sequencing

RNA was isolated from dissected E15.5 cerebral cortex (n = 5 $Phf6^{-/Y}$ and n = 3 $Phf6^{+/Y}$) and E16.5 cortical neurons cultured for 24 hours (as described above, n = 4 $Phf6^{lox/Y};Nes\text{-}cre^{Tg/+}$ and n = 3 $Phf6^{+/Y};Nes\text{-}cre^{Tg/+}$) was used for RNA sequencing. These tissues and cells were chosen, because E15.5 cortex is the developing cortex at peak neurogenesis directly taken ex vivo without any inference of culture systems, but with the caveat of a mixed population of cells, including some non-neuronal cells such as blood vessel endothelial cells. E16.5 neurons cultured overnight in a medium selects for neurons and provides a most homogeneous cell population of cortical neurons (>98% based on immunofluorescence marker staining). cDNA libraries were constructed using the TruSeq Stranded RNA Library Prep Kit (Illumina). The indexed libraries were pooled and diluted to 1.5 pM for paired-end 81bp sequencing on a NextSeq 500 instrument using the v2 150 Cycles High Output Kit (Illumina). The base calling and quality scoring were determined using Real-Time Analysis on board software v2.4.6, while the FASTQ file generation and de-multiplexing utilised bcl2fastq conversion software v2.15.04. All FASTQ files were aligned to the mouse genome, build mm10, using the Rsubread aligner v1.21.12 [88]. Reads overlapping each Entrez gene were summarized into counts using featureCounts [89] and Rsubread's inbuilt RefSeq annotation. Differential expression analyses were then undertaken independently for each cell type using the edgeR [90] and limma [91] software packages.

Gene annotation was downloaded from the NCBI (ftp.ncbi.nlm.nih.gov/gene/DATA/GEN-E_INFO). For each cell type all unknown, pseudo, and predicted genes were removed. Additionally, all ribosomal RNAs (rRNAs), genes with no official symbol and those not on a known chromosome were also removed. Following this, lowly expressed genes were filtered in each data set using edgeR's filterByExpr function with default settings. TMM normalisation [92] was then applied to the samples in each cell type and the data transformed to $\log_2$-counts per million reads (CPM).

Differential expression between the E15.5 cerebral cortex $Phf6^{-/Y}$ and $Phf6^{+/Y}$ samples, and the E16.5 cultured cortical neuron $Phf6^{lox/Y}Nes\text{-}cre^{Tg/+}$ and $Phf6^{+/Y}Nes\text{-}cre^{Tg/+}$ samples, was assessed using linear models and robust empirical Bayes moderated t-statistics with a trend prior variance [93]. P-values were adjusted to control the false discovery rate (FDR) below 5% using the Benjamini and Hochberg method. To increase precision, the linear models in each analysis incorporated a correction for a mouse litter effect.

The multi-dimensional scaling (MDS) plots, mean-difference (MD) plots and barcode plots were generated by limma's plotMDS, plotMD and barcodeplot functions respectively. Gene set tests were conducted using roast [94]. The annotations in org.Mm.eg.db_3.8.2 were used. Heatmaps were generated using the pheatmap package with row scaling and otherwise default settings.

## cDNA synthesis

The cDNA synthesis reaction mix consisted of 5 μM Oligo(dT)15 (Promega, Cat#C110A), 1 μl of dNTP mix (containing 10mM dATP, dGTP, dCTP and dTTP each, Bioline, BIO- 39044), RNA from E15.5 mouse cortex, and was made up to 20 μl with autoclaved $H_2O$. This mix was incubated at 65˚C for 5 min for primer annealing, then at 50˚C for 1 hour with 1 μl Super-Script III Reverse Transcriptase (Invitrogen, 18080085), 1 μl RnaseOut (Invitrogen, 10777019), 1 μl of 100 μM DTT (supplied with SuperScript III, Invitrogen, 18080085), and 4 μl of RT buffer (supplied with SuperScript III, Invitrogen, 18080085). To inactivate the enzyme, samples were incubated at 70˚C for 15 min, followed by incubation with Rnase H (Invitrogen, 100004927) to remove RNA.

## RT-qPCR

Each RT-qPCR reaction consisted of 10 μl of SensiMix SYBR Hi-ROX (Bioline, QT605), 6 μl H2O, 1 μl of each of primer (final 0.5 mM concentration), and 2 μl of cDNA. Each reaction was performed in technical triplicate. Samples were heated to 95˚C for 10 min in 384 well plates and then run through 40 cycles of: 95˚C for 20s, 60˚C for 20s and 72˚C for 30s using a QuantStudio 12K Flex (ThermoFisher). Standard curves for each primer pair were created using five-fold serial dilutions. Relative concentrations of the samples were obtained using the standard curve based on Cp values. Averages of each technical replicate were taken, then normalised to an average of the housekeeping genes (*Hsp90ab1*, *Pgk1*).

## WM-ISH probes and probe labelling

To generate cDNA templates, the mRNA sequence of the mouse gene of interest was obtained from UCSC online (http://genome.ucsc.edu/index.html) and submitted to Primer3 (http://bioinfo.ut.ee/primer3-0.4.0/primer3/) to generate primers producing an amplicon between 800 and 1200 bp in length, annealing at 60˚C. cDNA from a wild type C57BL/6 adult mouse brain was used to generate the amplicon via conventional PCR techniques, which was visualised by gel electrophoresis. The purified cDNA amplicon was cloned into a plasmid vector using the TOPO TA Cloning Kit for Sequencing (Invitrogen, 450030), with OneShot competent cells according to manufacturer's instructions. Plasmid DNA was extracted using the PureLink Quick Plasmid Miniprep Kit (Life Technologies, K210010). A sample of each clone was checked by PCR and capillary sequencing. DIG-labelled cRNA probes were generated using the DIG RNA Labelling Kit (Roche, 11175025910) as per manufacturer's protocol.

## Whole-mount in-situ-hybridisation (WM-ISH)

Whole E15.5 mouse brains were harvested and fixed in 4% (w/v) PFA/PBS for 2 h at 4˚C with gentle rocking. Fixed brains were rinsed once in PBST (PBS + 0.1% Tween20; Sigma, P7949) before 5 min incubations in a dehydration series of 25%, 50%, 75% (v/v) methanol/PBS and 100% methanol. *Day 1*: brains were serially rehydrated by incubating for 5 min each in 75%, 50%, 25% (v/v) methanol PBST and twice in PBST. Brains were incubated for 60 min in 6% $H_2O_2$ in PBST at RT to block endogenous peroxidases and bleach the tissue, washed in PBST (3x 5 min at RT) and permeabilised for 20 min in10 μg/ml proteinase K in PBST. Proteinase K was quenched by incubating brains in 2 mg/ml glycine in PBST for 5 min then washed in PBST (2x 5 min at RT). Brains were refixed in 0.2% (v/v) glutaraldehyde in 4% (w/v) PFA/PBS for 20 min, then washed in PBST (2x 5 min at RT), before incubating in pre-hybridisation buffer (50% (v/v) deionised formamide, 5x SSC pH 4.5, 1% (w/v) SDS, 50 μg/ml yeast RNA, 50 μg/ml heparin in MQ-H2O) at 65˚C in a hybridisation oven for 1h. 10 μl of antisense probe

against *Nefl*, *Peg10*, *Phip* and *Zdbf2* were denatured for 5 min at 95˚C, then added to 1 ml pre-heated hybridisation buffer. The brains were incubated in the probe/hybridisation buffer mix overnight at 65˚C in a hybridisation oven with gentle rotation. *Day 2*: Solution 1 (50% form-amide, 5x SSC pH 4.5, 1% SDS in MQ-$H_2O$), 1:1 Solution 1:Solution 2 (0.5 M NaCl, 10 mM Tris-HCl, 0.1% Tween-20 in MQ-$H_2O$) and Solution 3 (50% formamide, 2x SSC pH 4.5 in MQ-$H_2O$) were prepared. Brains were rinsed once in Solution 1, before being incubated in Solution 1 (2x 30 min), then 10 min in 1:1 Solution 1:Solution 2. At room temperature, brains were washed in RT Solution 2 (3x 5 min), before incubating at 37˚C in 100 µg/ml RnaseA in Solution 2 (2x 30 min) followed by 5 min wash at RT in Solution 2. Brains were returned to the 65˚C hybridisation oven in Solution 3 (2x 30 min), then RT $MAB^L$ (100 mM maleic acid, 150 mM NaCl in MQ-$H_2O$ pH 7.5 with 0.48 mg/ml Levamisole) (1 x 15 min), 2% BMB (Block-ing Reagent; 11096176001, Roche) in $MAB^L$ (1 x 15 min) and 20% foetal calf serum, 2% BMB in $MAB^L$ (1 x 1 h). Brains were then incubated overnight at 4˚C in 1:2000 alkaline phospha-tase-conjugated anti-Digoxigenin-antibody-AP, Fab Fragments (11093 274 910, Roche) pre-blocked at 4˚C in 20% FCS, 2% BMB in $MAB^L$. *Day 3*: All steps were performed at RT. Brains were washed in $MAB^L$ (3x 5 min) then in $MAB^L$ (4x 1 h) before incubating overnight at RT in $NTMT^L$ (100 mM NaCl, 100 mM Tris-HCl pH 9.5, 50 mM $MgCl_2$, 1% Tween-20, 2 mM Levamisole in MQ-$H_2O$). *Day 4*: All steps were performed at RT, unless otherwise stated. Brains were washed in $NTMT^L$ (3x 10 min) followed by addition of 20 µl/ml alkaline phospha-tase substrate NBT-BCIP (11681541001, Roche) in $NTMT^L$. The signal was developed at 4˚C at least overnight. The reaction was stopped by removing substrate solution and washing the brains in PBST for 5 min, followed by incubating at RT in PBST (2x 30 min), then clearing at RT in 40% glycerol (4˚C O/N), 60% glycerol (4˚C O/N) then 80% glycerol for photography.

## Calcium imaging

E16.5 cortical neurons were cultured as described above. At day 5, the culture medium was removed, and cells were incubated with Fluo-4 AM (Invitrogen, F14217) in complete neuroba-sal differentiating medium for 1 h at 37˚C and 5% $CO^2$. Cells were then washed with complete neurobasal differentiating medium without dye (3x 5 min). Fresh complete neurobasal differ-entiating medium was added and cells were imaged with a live-cell widefield inverted micro-scope (Axio Observer, Zeiss). For a single excitation wavelength of Fluo-4, the excitation light was filtered through a 460–490 nm band pass filter and the emission light passed through a 515–550 nm band pass filter and captured by a high-speed digital camera. Acquisition proto-cols consisted of 5-minute time-lapse sequences of Fluo-4 fluorescence at 100 msec intervals. Analysis, processing and playback of the image sequences for visual inspection was conducted using *ImageJ* /FIJI software ((http://fiji.sc/Fiji). To assess differences in calcium resulting from spontaneous activity, the raw image sequences were processed to highlight changes in fluores-cence intensity between frames. All cells were assessed as regions of interest and the mean pixel intensity at each frame was measured. The data were plotted as fluorescence intensity versus time (z profile) and subsequently converted to a relative scale (ΔF/F baseline) and a 0 to 1 range. Peaks and valleys were identified using a peak and valley detection algorithm devised by Tom O'Haver (https://terpconnect.umd.edu/~toh/spectrum/PeakFindingandMeasurement.htm#Spreadsheet). The following cut-off for background, minor and major peaks were used: (1) Peaks with a height of less than 1% of the maximal peak amplitude were considered background and excluded from the analysis. (2) Peaks with a height larger than 1% and less than 10% of the maximal peak amplitude were considered minor peaks. (3) Peaks with a height larger than 10% of the maximal peak amplitude were con-sidered major peaks.

### BFLS patient survey

An online CoRDS survey for BFLS patients as part of the BFLS Inc group has been established. The goal is to have both a registry and collect information for a natural history study of BFLS. BFLS patient families who are interested in participating can access the survey using this link: https://cords.sanfordresearch.org/activation-form.

## Supporting information

**S1 Table.** *PHF6* **mutations and references for the BFLS samples shown in Fig 1.** (PDF)

**S2 Table. Effect of** *Phf6* **germline mutation on survival on the C57BL/6 background.** (PDF)

**S3 Table. RNA-sequencing results of 5** $Phf6^{-/Y}$ ***vs.*** **3** $Phf6^{+/Y}$ **E15.5 cortices. Tab A in S3 Table**: Genes differentially expressed in $Phf6^{-/Y}$ vs. $Phf6^{+/Y}$ cortices. **Tab B in S3 Table**: KEGG pathways enriched among differentially expressed genes in $Phf6^{-/Y}$ vs. $Phf6^{+/Y}$ cortices. **Tab C in S3 Table**: GO pathways enriched in genes expressed in $Phf6^{-/Y}$ vs. $Phf6^{+/Y}$ cortices. **Tab D in S3 Table**: List of gene differentially expressed in $Phf6^{-/Y}$ vs. $Phf6^{+/Y}$ *cortices* required for brain, cortex and neuronal development. (XLSX)

**S4 Table. RNA-sequencing results of cortical neurons from 4** $Phf6^{lox/Y};Nes-cre^{Tg/+}$ **vs. 3** $Phf6^{+/Y};Nes-cre^{Tg/+}$ **E16.5 foetuses. Tab A in S4 Table**: Genes differentially expressed in $Phf6^{lox/Y};Nes-cre^{Tg/+}$ *vs.* $Phf6^{+/Y};Nes-cre^{Tg/+}$ cortical neurons. **Tab B in S4 Table**: KEGG pathways enriched among differentially expressed genes in $Phf6^{lox/Y};Nes-cre^{Tg/+}$ *vs.* $Phf6^{+/Y};Nes-cre^{Tg/+}$ cortical neurons. **Tab C in S4 Table**: GO pathways enriched in genes expressed in $Phf6^{lox/Y};Nes-cre^{Tg/+}$ *vs.* $Phf6^{+/Y};Nes-cre^{Tg/+}$ cortical neurons. **Tab D in S4 Table**: List of gene differentially expressed in $Phf6^{lox/Y};Nes-cre^{Tg/+}$ *vs.* $Phf6^{+/Y};Nes-cre^{Tg/+}$ cortical neurons and required for brain, cortex and neuronal development. (XLSX)

**S1 Data. Source data underlying Figs 1–5, 7, S1, S2, S4, S6, S7, S8, S10, S11, S14 and S15.** (XLSX)

**S1 Fig. Females with homozygous** *Phf6* **deletion in the central nervous system are susceptible to seizures.** Time to first seizure in a small number of females with homozygous deletion of *Phf6* in the nervous system. The median latency was 400 days. Controls are the same cohort shown in Fig 2B. n = 11 $Phf6^{+/+};Nes-cre^{Tg/+}$ mice, n = 4 $Phf6^{lox/lox};Nes-cre^{Tg/+}$ mice. Data were analysed by a log-rank (Mantel-Cox) test. (PDF)

**S2 Fig. No change in GAD67$^+$ neurons in** *Phf6*-**deleted mice.** (A) Representative sections of N = 3 $Phf6^{+/Y};Nes-cre^{Tg/+}$ and 3 $Phf6^{lox/Y};Nes-cre^{Tg/+}$ adult brains (13–14 weeks old) stained with anti-GAD67 antibody by immunohistochemistry. Scale bar equals 100 μm. The schematic on the right indicates the approximate locations where GAD67$^+$ cells were quantified in sections in the left and right hemisphere on each cortex section. (B) Enumeration of GAD67$^+$ cells per mm$^2$ of cortex section. The average ± sem for each genotype is shown with circles representing data from individual animals. The number of GAD67$^+$ cells was counted on matched sections of the parietal cortex and divided by the area analysed. Data were analysed by two-tailed Student's t-test (p = 0.51). (PDF)

**S3 Fig. Histology of adult seizure-affected $Phf6^{lox/Y}$;$Nes$-$cre^{Tg/+}$ and $Phf6^{+/Y}$;$Nes$-$cre^{Tg/+}$ control brains.** Representative images showing matched 10 μm brain sections of a $Phf6^{lox/Y}$;$Nes$-$cre^{Tg/+}$ and a $Phf6^{+/Y}$;$Nes$-$cre^{Tg/+}$ brain stained with cresyl violet. Serial sections of N = 3 $Phf6^{lox/Y}$;$Nes$-$cre^{Tg/+}$ mice 17–37 days after onset of seizures and 3 age-matched $Phf6^{+/Y}$;$Nes$-$cre^{Tg/+}$ control mice (394 to 483 days old) were examined. Scale bar = 1 mm.
(PDF)

**S4 Fig. The number of Purkinje cells of the cerebellum is not affected by loss of PHF6.** (A) Representative image of the cerebellum of N = 3 $Phf6^{+/Y}$;$Nes$-$cre^{Tg/+}$ and 3 $Phf6^{lox/Y}$;$Nes$-$cre^{Tg/+}$ brain. Scale bar = 200 μm. (B) Quantification of the number of Purkinje cells in the cerebellum in three mice per genotype (aged 394 to 483 days old). Cells were counted in 4 evenly spaced sections per animal. Data are presented as the number of Purkinje cells per length of the Purkinje cell layer in mm showing the results from individual animals as a circle and the mean ± sem for each genotype. Data were analysed using a two-tailed Student's t-test (p = 0.17).
(PDF)

**S5 Fig. Reduction in cortex volume and other regions of the brain relative to brain volume caused by loss of PHF6.** (A) Volume of the cerebral cortex divided by the volume of the total brain in the sections surrounding the lateral ventricles is slightly reduced in $Phf6^{lox/Y}$;$Nes$-$cre^{Tg/+}$ vs. $Phf6^{+/Y}$;$Nes$-$cre^{Tg/+}$ brains. (B) Volume of the rest of the brain (i.e., minus the cortex and lateral ventricles), divided by the volume of the total brain in the sections surrounding the lateral ventricles in $Phf6^{lox/Y}$;$Nes$-$cre^{Tg/+}$ vs. $Phf6^{+/Y}$;$Nes$-$cre^{Tg/+}$ brains. LV = lateral ventricles. (C) Depth of individual cortical layers as percentage of total cortex depth in $Phf6^{lox/Y}$;$Nes$-$cre^{Tg/+}$ vs. $Phf6^{+/Y}$;$Nes$-$cre^{Tg/+}$ brains. N = 3 $Phf6^{lox/Y}$;$Nes$-$cre^{Tg/+}$ vs. 3 $Phf6^{+/Y}$;$Nes$-$cre^{Tg/+}$ mice (aged 394 to 483 days old). Circles represent individual animals (A-C). Lines between individual data points join each $Phf6^{lox/Y}$;$Nes$-$cre^{Tg/+}$ animal to its paired control (A,B). Data (C) are presented as mean ± sem. Data were analyzed by a paired t-test (A,B) or unpaired two-tailed t-test (C).
(PDF)

**S6 Fig. Assessment of ventricle area in brain sections of 13 to 14-week-old mice prior to the onset of seizures.** (A) Representative images of cresyl violet stained paraffin section of $Phf6^{lox/Y}$;$Nes$-$cre^{Tg/+}$ vs. $Phf6^{+/Y}$;$Nes$-$cre^{Tg/+}$ 13–14-week-old mouse brains at the level of the parietal cortex. Scale bar = 1 mm. (B) Assessment of the total brain area and the relative area of the lateral ventricles at the level of the frontal cortex and the 3rd ventricle at the level of the parietal cortex. The ventricle area is expressed as percentage of the total brain tissue area in the same section. Four sections per animal, two per brain region at the level of the frontal and parietal cortex were assessed. The left and right lateral ventricle were assessed. N = N = 3 $Phf6^{lox/Y}$;$Nes$-$cre^{Tg/+}$ vs. 3 $Phf6^{+/Y}$;$Nes$-$cre^{Tg/+}$ 13–14-week-old mice. Data are displayed as mean ± sem and were analysed by two-way ANOVA (B, left) and unpaired, two-tailed Student's t test (B, middle and right).
(PDF)

**S7 Fig. Effect of $Phf6$ mutation on the incidence hydrocephalus with enlarged skull.** (A) Incidence of hydrocephalus with enlarged skull of N = 205 female $Phf6^{+/+}$ and 268 female $Phf6^{+/-}$ mice. (B) Incidence of hydrocephalus with enlarged skull of N = 104 female $Phf6^{+/+}$;$Nes$-$cre^{Tg/+}$ and 101 female $Phf6^{+/lox}$;$Nes$-$cre^{Tg/+}$. No significant difference was detected. (C) Incidence of hydrocephalus with enlarged skull of N = 109 male $Phf6^{+/Y}$;$Nes$-$cre^{Tg/+}$ and 74 male $Phf6^{lox/Y}$;$Nes$-$cre^{Tg/+}$. No significant difference was detected. Checked points indicate mice that were euthanized for reasons other than hydrocephalus. Data were analysed by

Mantel-Cox log-rank test.
(PDF)

**S8 Fig. Cortical layering is unaffected by heterozygous loss of *Phf6*.** (A) Percentage of cells expressing each cortical layer marker protein (SATB2, CTIP2, TBR1) in each of 10 pial to ventricular bins on parietal sections of three mice per genotype. No significant differences were detected between genotypes. (B) Percentage of cells for each cell type across the parietal cortex sections analysed. No significant differences were detected between genotypes. Circles represent individual mouse foetuses. N = 3 *Phf6*$^{+/+}$ and 3 *Phf6*$^{+/-}$ foetuses. Data are presented as mean ± sem and were analysed by a two-tailed Student's t-test.
(PDF)

**S9 Fig. Cortical layering of the adult cortex is unaffected by loss of *Phf6*.** Representative images of immunohistochemistry staining for reelin, CUX1 (layers II-V), CTIP2 (layers V-VI) and FOXP2 (layer VI) of sections of the parietal cortex of 13-14-week-old *Phf6*$^{lox/Y}$;*Nes-cre*$^{Tg/+}$ vs. *Phf6*$^{+/Y}$;*Nes-cre*$^{Tg/+}$ mice. N = 3 mice per genotype. Scale bar = 180 μm.
(PDF)

**S10 Fig. Neurite length and number of cultured cortical neurons are unaffected by loss of PHF6.** (A) Representative images from E16.5 *Phf6*$^{lox/Y}$;*Nes-cre*$^{Tg/+}$ vs. *Phf6*$^{+/Y}$;*Nes-cre*$^{Tg/+}$ cultured cortical neurons showing staining with anti-beta III tubulin and DAPI. Scale bar = 21.25 μm. (B) Enumerations of the numbers of primary, secondary and tertiary neurites per neuron in control and *Phf6*-deleted cultured cortical neurons. No significant difference was detected. (C) Quantification of the length of primary, secondary and tertiary neurites in *Phf6*$^{lox/Y}$;*Nes-cre*$^{Tg/+}$ vs. *Phf6*$^{+/Y}$;*Nes-cre*$^{Tg/+}$ cultured cortical neurons. No significant difference was detected. N = 4 *Phf6*$^{lox/Y}$;*Nes-cre*$^{Tg/+}$ vs. 4 *Phf6*$^{+/Y}$;*Nes-cre*$^{Tg/+}$ foetuses. Data are displayed mean ± sem. Circles represent individual mouse foetuses. Data were analysed by using a two-tailed Student's t-test.
(PDF)

**S11 Fig. Loss of PHF6 does not affect neural stem and progenitor proliferation.** (A,B) Cell proliferation parameters from passage 0 (primary isolation and plating) to passage 13 of neural stem and progenitor cells (NSPCs) isolated from the dorsal telencephalon of E12.5 *Phf6*$^{lox/Y}$; *Nes-cre*$^{Tg/+}$ vs. *Phf6*$^{+/Y}$;*Nes-cre*$^{Tg/+}$ embryos. (A) Cell yield at each consecutive passage after plating 50,000 cells per well of 6-well tissue culture dishes. (B) Cumulative cell growth. N = NSPC isolates from 8 embryos per genotype. Data are presented as mean ± sem (A,B). Circles represent cells isolated from individual embryos (A). Data were analysed by two-way-ANOVA with genotype and passage number as the two independent factors.
(PDF)

**S12 Fig. Flow cytometry gating strategy to assess cell type markers in differentiating neural stem and progenitor cells.** (A,B) Flow cytometry scatter plots of representative *Phf6*$^{+/Y}$;*Nes-cre*$^{Tg/+}$ (A) and *Phf6*$^{lox/Y}$;*Nes-cre*$^{Tg/+}$ (B) neural stem and progenitor cells (NSPCs) grown in differentiation medium (without FGF2 or EGF and with 1% FCS) for 5 days. Side scatter (SSC) and forward scatter (FSC) width (W) and height (H) were used to exclude cell doublets; SSC and FSC areas (A) were used to exclude cell debris; a fixable live/cell death marker was used to select live cells; fluorescently labelled antibodies (see methods) were used to detect astrocytes [glial fibrillary acidic protein positive cells (GFAP$^{+}$) and S100 protein, beta polypeptide, neural positive (S100ß$^{+}$)], oligodendrocytes (monoclonal antibody O4$^{+}$) and neurons (ßIII-tubulin$^{+}$).
(PDF)

**S13 Fig. RNA-sequencing analysis of *Phf6*-deleted and control E15.5 cortex and E16.5 cultured cortical neurons.** (A) Multidimensional scaling (MDS) plot of N = 5 $Phf6^{-/Y}$ *vs.* 3 $Phf6^{+/Y}$ E15.5 cortices samples used for RNA-sequencing analysis, showing clustering of samples after adjustment for litter-effect; color-coded by genotype. (B) MDS plot of N = 4 $Phf6^{lox/Y};Nes\text{-}cre^{Tg/+}$ vs. 3 $Phf6^{+/Y};Nes\text{-}cre^{Tg/+}$ E16.5 cortical neuron isolates used for RNA-sequencing analysis showing clustering of samples after adjustment for litter-effect; color-coded by genotype. (C,D) Mean-difference plot for $Phf6^{-/Y}$ *vs.* $Phf6^{+/Y}$ E15.5 cortex RNA-sequencing data showing $\log_2$-fold change for each gene over average $\log_2$ expression (C). Mean-difference plot for $Phf6^{lox/Y};Nes\text{-}cre^{Tg/+}$ vs. $Phf6^{+/Y};Nes\text{-}cre^{Tg/+}$ E16.5 cultured cortical neuron RNA-sequencing data (D). Significantly up- and downregulated genes in the *Phf6* deleted samples relative to controls are coloured red and blue respectively. NS = not significant. (E) $\log_2$ fold-change in RNA levels of ZIC family transcription factor genes in E15.5 $Phf6^{-/Y}$ *vs.* $Phf6^{+/Y}$ cortex. (F) $\log_2$ fold-change in RNA levels of the *Usp18* gene in $Phf6^{lox/Y};Nes\text{-}cre^{Tg/+}$ vs. $Phf6^{+/Y};Nes\text{-}cre^{Tg/+}$ E16.5 cortical neurons. RNA sequencing data were analysed as described in the methods section.
(PDF)

**S14 Fig. Additional assessment of the expression of selected genes identified as differentially expressed between Phf6 deleted and control samples by RNA-sequencing.** (A) RT-qPCR of $Phf6^{lox/Y};Nes\text{-}cre^{Tg/+}$ vs. $Phf6^{+/Y};Nes\text{-}cre^{Tg/+}$ E15.5 cortex. As in the RNA-sequencing dataset, mRNA levels for the selected genes were higher in the $Phf6^{lox/Y};Nes\text{-}cre^{Tg/+}$ vs. $Phf6^{+/Y};Nes\text{-}cre^{Tg/+}$ cortex. (B) Representative images of RNA/RNA whole mount in situ hybridisation of E15.5 $Phf6^{lox/Y};Nes\text{-}cre^{Tg/+}$ vs. $Phf6^{+/Y};Nes\text{-}cre^{Tg/+}$ brains using RNA probes as indicated. N = 5–7 tissue samples per genotype. Data are presented as mean ± sem (A). Data from individual foetuses are represented as open circles (A). Data were analysed by two-way ANOVA (A).
(PDF)

**S15 Fig. $Phf6^{+/Y};Nes\text{-}cre^{Tg/+}$ cortical neurons display a higher number of calcium signalling peaks per foetus than $Phf6^{lox/Y};Nes\text{-}cre^{Tg/+}$ cortical neurons.** (A,B) Calcium imaging results from 906 $Phf6^{+/Y};Nes\text{-}cre^{Tg/+}$ and 932 $Phf6^{lox/Y};Nes\text{-}cre^{Tg/+}$ cortical neurons from N = 8 $Phf6^{+/Y};Nes\text{-}cre^{Tg/+}$ and 4 $Phf6^{lox/Y};Nes\text{-}cre^{Tg/+}$ E16.5 foetuses. Number of peaks above background per cell per foetus in six peak height amplitude bins from 0 to <30% of total peak height (A) and 30% to maximum (B). Data are displayed as mean ± sem. Circles represent data from individual foetuses. Data were analysed by two-way ANOVA.
(PDF)

## Acknowledgments

We thank F. Dabrowski and WEHI Bioservices for assistance with animal experiments, M. Dixon for creating the *Phf6* targeted construct and J. Chue for preliminary work on the *Phf6* mutant mice. We thank R.E. May, S. Eccles and H. Pehlivanoglu for excellent technical support. We thank J. Crawford for her help with the patient cell lines and technical optimization of the anti-PHF6 antibody.

## Author Contributions

**Conceptualization:** Anthony J. Hannan, Jozef Gécz, Tim Thomas, Anne K. Voss.

**Data curation:** Alexandra L. Garnham, Yifang Hu, Gordon K. Smyth.

**Formal analysis:** Helen M. McRae, Melody P. Y. Leong, Maria I. Bergamasco, Alexandra L. Garnham, Yifang Hu, Lachlan Whitehead, Gordon K. Smyth, Anne K. Voss.

**Funding acquisition:** Jozef Gécz, Tim Thomas, Anne K. Voss.

**Investigation:** Helen M. McRae, Melody P. Y. Leong, Maria I. Bergamasco, Mark A. Corbett, Farrah El-Saafin, Bilal N. Sheikh, Stephen Wilcox.

**Project administration:** Anne K. Voss.

**Software:** Alexandra L. Garnham.

**Supervision:** Jozef Gécz, Gordon K. Smyth, Tim Thomas, Anne K. Voss.

**Validation:** Helen M. McRae, Melody P. Y. Leong, Maria I. Bergamasco, Mark A. Corbett.

**Visualization:** Helen M. McRae, Melody P. Y. Leong, Maria I. Bergamasco, Alexandra L. Garnham, Mark A. Corbett, Anne K. Voss.

**Writing – original draft:** Helen M. McRae, Melody P. Y. Leong, Maria I. Bergamasco, Alexandra L. Garnham, Anne K. Voss.

**Writing – review & editing:** Mark A. Corbett, Anthony J. Hannan, Jozef Gécz, Gordon K. Smyth, Tim Thomas, Anne K. Voss.

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
