## [Decision Letter · Decision Letter 0]

6 Aug 2024

Dear Dr Voss,

Thank you very much for submitting your Research Article entitled 'Loss of PHF6 causes spontaneous seizures, enlarged brain ventricles and altered transcription in the cortex of a mouse model of the Börjeson–Forssman–Lehmann intellectual disability syndrome' to PLOS Genetics.

The manuscript was fully evaluated at the editorial level and by independent peer reviewers. The reviewers appreciated the attention to an important topic but identified some concerns that we ask you address in a revised manuscript.

We therefore ask you to modify the manuscript according to the review recommendations. Your revisions should address the specific points made by each reviewer.

To resubmit, log into your Editorial Manager account and select the option 'Revise Submission' in the 'Submissions Needing Revision' folder.

Yours sincerely,

Hongyan Wang, Ph.D.

Academic Editor

PLOS Genetics

Gregory Barsh

Section Editor

PLOS Genetics

Reviewer's Responses to Questions

**Comments to the Authors:**

Reviewer #1: In this manuscript, McRae et al. use conditional knockout mice to investigate the phenotype caused by the loss of the Phf6 gene in the mouse brain. This study helps to better understand the impact of PHF6 genetic mutations on human brain development and the pathological defects of Börjeson-Forssman-Lehmann syndrome. The authors have made improvements in the revision; however, the article still needs better organization and further analyses to make the conclusions more reliable. Some comments are listed below with the hope that the authors will find them useful.

1. In Fig 1A, why does it show that the Ponceau S staining bands were even but beta-tubulin bands were not? This needs replicates and quantifications.

2. In Fig 2, the quantification is about the “Time to first seizure in female mice.” How about the frequency after the first seizure?

3. One of the most confusing points is that the authors claimed “Phf lox/Y;Nes-creTg/+ NSPCs formed more neurons (ßIII-tubulin+) and fewer astrocytes (GFAP+, S100ß) (Lin291).” But the cortex volume reduced (Fig 3E) with no layer neuron increase in the Phf6+/- mouse cortex (Fig S8AB). How can this be explained?

4. In Fig 6A,B, please add gene names in the graph. In addition, it is very surprising that only 9-10 genes were downregulated and 30-50 genes were upregulated. Only using 30-50 genes for GO analysis in Fig 6G and H is not a reliable way to get a solid conclusion. How many genes are in each pathway? Do these DEGs play a truly essential role in these pathways?

5. Line 373, how are minor and major peaks defined? How does this relate to gene expression profiles and link to seizures?

Reviewer #2: BFLS is an X-linked intellectual disability and endocrine disorder, caused by mutations in the PHF6 gene. To understand the pathogenesis of PHF6 mutations in BFLS, the authors used PBMCs from patients carrying PHF6 variants to examine PHF6 protein levels. Based on the patient results, they have generated two PHF6 mice lines: loss of Phf6 in the germline and CNS-specific deletion of Phf6. They further characterized phenotypic, anatomical, cellular and molecular changes in these mice. They found that cerebral cortex is the site of higher brain functions for cognition and decision-making. Loss of PHF6 results in the dysregulation of neuronal development and differentiation genes. Lacking of PHF6 in mice recapitulates BFLS patients in spontaneous epileptic seizures. Overall, the mice models and findings are useful in understanding the role of PHF6 in the pathogenesis of BFLS.

Although there is a long gap of resubmission due to Covid-19, the authors had addressed most of the reviewer’s concerns and the manuscript significantly improved by adding new data and revision.

**Have all data underlying the figures and results presented in the manuscript been provided?**

Reviewer #1: Yes

Reviewer #2: Yes

PLOS authors have the option to publish the peer review history of their article (what does this mean?). If published, this will include your full peer review and any attached files.

Reviewer #1: No

Reviewer #2: No

---

## [Decision Letter · Decision Letter 1]

11 Sep 2024

Dear Dr Voss,

We are pleased to inform you that your manuscript entitled "Loss of PHF6 causes spontaneous seizures, enlarged brain ventricles and altered transcription in the cortex of a mouse model of the Börjeson–Forssman–Lehmann intellectual disability syndrome" has been editorially accepted for publication in PLOS Genetics. Congratulations!

Yours sincerely,

Hongyan Wang, Ph.D.

Academic Editor

PLOS Genetics

Gregory Barsh

Section Editor

PLOS Genetics

Comments from the reviewers (if applicable):

Reviewer's Responses to Questions

**Comments to the Authors:**

Reviewer #1: The authors have addressed my comments.

**Have all data underlying the figures and results presented in the manuscript been provided?**

Reviewer #1: Yes

PLOS authors have the option to publish the peer review history of their article (what does this mean?). If published, this will include your full peer review and any attached files.

Reviewer #1: No

**Data Deposition**

http://datadryad.org/submit?journalID=pgenetics&manu=PGENETICS-D-24-00726R1

**Press Queries**

---

## [Editor Report · Acceptance letter]

30 Sep 2024

PGENETICS-D-24-00726R1 

Loss of PHF6 causes spontaneous seizures, enlarged brain ventricles and altered transcription in the cortex of a mouse model of the Börjeson–Forssman–Lehmann intellectual disability syndrome 

Dear Dr Voss, 

We are pleased to inform you that your manuscript entitled "Loss of PHF6 causes spontaneous seizures, enlarged brain ventricles and altered transcription in the cortex of a mouse model of the Börjeson–Forssman–Lehmann intellectual disability syndrome" has been formally accepted for publication in PLOS Genetics! Your manuscript is now with our production department and you will be notified of the publication date in due course.

With kind regards,

Lilla Horvath

PLOS Genetics

On behalf of:
